# The Klarna Product Page Dataset: Web Element Nomination with Graph Neural Networks and Large Language Models

**Alexandra Hotti**                                          *alexandrahotti@gmail.com*
*KTH Royal Institute of Technology, School of EECS*
*Klarna*

**Riccardo S. Risuleo**                                      *riccardosven@gmail.com*
*Klarna*

**Stefan Magureanu**                                         *smagureanu@gmail.com*
*Klarna*

**Aref Moradi**                                              *rfmrd619@gmail.com*
*Klarna*

**Jens Lagergren**                                           *jens.lagergren@scilifelab.se*
*KTH Royal Institute of Technology, School of EECS*
*SciLifeLab*

**Reviewed on OpenReview:** *https://openreview.net/forum?id=zz6FesdDbB*

## Abstract

Web automation holds the potential to revolutionize how users interact with the digital world, offering unparalleled assistance and simplifying tasks via sophisticated computational methods. Central to this evolution is the web element nomination task, which entails identifying unique elements on webpages. Unfortunately, the development of algorithmic designs for web automation is hampered by the scarcity of comprehensive and realistic datasets that reflect the complexity faced by real-world applications on the Web. To address this, we introduce the Klarna Product Page Dataset, a comprehensive and diverse collection of webpages that surpasses existing datasets in richness and variety. The dataset features 51,701 manually labeled product pages from 8,175 e-commerce websites across eight geographic regions, accompanied by a dataset of rendered page screenshots. To initiate research on the Klarna Product Page Dataset, we empirically benchmark a range of Graph Neural Networks (GNNs) on the web element nomination task. We make three important contributions. First, we found that a simple Convolutional GNN (GCN) outperforms complex state-of-the-art nomination methods, and further enhance its performance using a Reversible GNN (RevGNN) architecture. Second, we introduce a training refinement procedure that involves identifying a small number of relevant elements from each page using the aforementioned GNN. These elements are then passed to a Large Language Model for the final nomination. This procedure significantly improves the nomination accuracy by 10.9 percentage points on our challenging dataset, without any need for fine-tuning. Finally, in response to another prevalent challenge in this field – the abundance of training methodologies suitable for element nomination – we introduce the *Challenge Nomination Training Procedure*, a training method that further boosts nomination accuracy.

## 1 Introduction

In the rapidly advancing field of web automation (Yao et al., 2022; Lin et al., 2020; Wang et al., 2022; Deng et al., 2023), an overarching goal is to provide users with seamless assistance when completing mundane

tasks online. By harnessing the full potential of web automation, users will no longer be forced to manually navigate through tedious workflows. Instead, intrusive pop-ups will be preemptively closed, and forms will be automatically completed, regardless of the specific website being accessed. Automation features like these could significantly enhance the user experience in, for example, super-apps equipped with integrated browsers that require advanced automation capabilities. Apart from saving users' time and attention, web automation paves the way for creating personal online assistants. These assistants could, for example, allow a user to purchase a specific pair of shoes at the best possible price with just a simple instruction.

One factor driving the recent heightened interest in web automation is the rise of Large Language Models (LLMs). The impressive performance demonstrated by LLMs, combined with the fact that models such as GPT-4 (OpenAI et al., 2023) have been trained on HTML data, has catalyzed a surge in research focused on using LLMs as, for instance, foundation models for navigation on webpage datasets (Furuta et al., 2023; Zhao et al., 2023; Deng et al., 2023; Zheng et al., 2024; Wang et al., 2023; Kim et al., 2023; Zheng et al., 2023; Gur et al., 2023).

Beyond navigation capabilities, many web automation features also require the ability to precisely locate information on a page. For instance, when purchasing the aforementioned shoes for a user, it is necessary to extract and compare prices from product detail pages across a large number of vendors. Additionally, as Deng et al. (2023) highlight, automation capabilities could be integrated as plugins for models like ChatGPT, enabling on the fly retrieval of information from HTML websites based on user inputs, instead of relying on predefined APIs for each web service. Clearly, many web automation features require a combination of navigation and information retrieval capabilities. However, as far as we are aware, few, if any, studies or datasets comprehensively address both aspects.

The scarcity of publicly accessible, large-scale, and realistic webpage datasets is a significant barrier to progress in this field. Previous research has been limited to simplified, simulated webpages (Gur et al., 2023; Yao et al., 2023; Shi et al., 2017a), or confined to pages from a narrow selection of websites (Hao, 2011; Yao et al., 2023). Such datasets fail to reflect the complexities faced by real applications on the Web, which consists of tens of millions of websites, each with thousands of elements and layout patterns that vary significantly across different sites. This shortage may stem from the fact that collecting and annotating a comprehensive web page dataset is a costly and time-consuming undertaking, requiring skills more commonly found among web developers than machine learning researchers.

As a significant contribution to the community, we introduce the *Klarna Product Page Dataset*[1]. Figure 10 presents screenshots of four rendered pages from our dataset. The Klarna Product Page Dataset is designed to emulate a scenario encountered by, for instance, an autonomous shopping assistant. This assistant is tasked with navigating through product pages across a multitude of different websites to extract product information. The goal could be to identify a specific item requested by a user at a given price point, then add this item to the cart and proceed to the checkout page. However, note that in the empirical evaluation carried out in this work, we solely focus on identifying the elements individually.

Furthermore, the Klarna Product Page Dataset dataset comprises over $50,000$ real e-commerce pages from $8,175$ websites, with five manually labeled action and information elements per page (See Figure 2). To the best of our knowledge, this positions our dataset as the largest available labeled webpage dataset in terms of the number of websites. Additionally, while the majority of public datasets primarily consist of English pages, a unique advantage of Klarna Product Page Dataset is its inclusion of pages from eight distinct geographic markets. This inclusion enables the capability to assess generalizability across multiple languages (See Table 1). Finally, recent studies have also leveraged webpage screenshots to explore computer vision research for the web (Zheng et al., 2024; Kumar et al., 2022). To support future similar efforts, we have developed a complementary dataset that includes screenshots of the rendered pages from our dataset.

As a starting point for research on the *Klarna Product Page Dataset*, we benchmark the performance of various neural networks on *Web element nomination*, which is a fundamental concept for automating tasks on the web. We define Web element nomination as the process of identifying a unique element from a specific class on a webpage. One of the challenges that makes this task particularly demanding is that among potentially

---

[1] Available at github.com/klarna/product-page-dataset under the CC BY-NC-SA license.

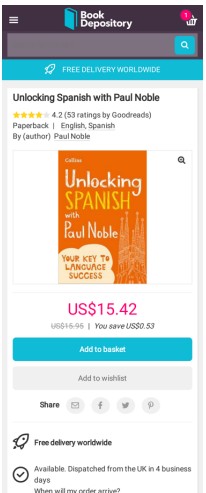 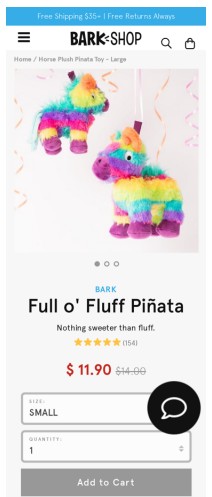 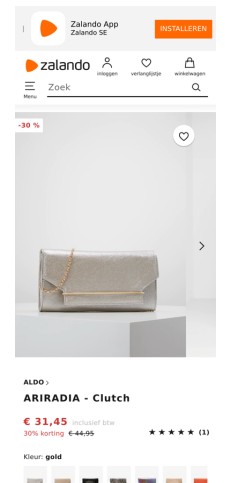 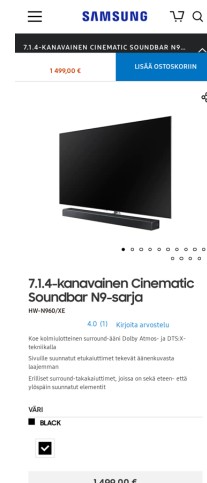

Figure 1: Four pages from the Finnish, Dutch and US markets in our dataset.

thousands of nodes on a page, only one element belongs to each class. For this purpose, we train several neural networks on the Document Object Model (DOM) tree representations of webpages, using three popular families of graph neural networks (GNNs): recurrent GNNs (Tree Long Short-Term Memory networks), convolutional GNNs (GCNs), and attention-based GNNs. Our aim is to compare their performance in web element nomination and identify the most promising GNN family for future research on the *Klarna Product Page Dataset*. We also compare their performance against state-of-the-art (SOTA) methods FreeDOM (Lin et al., 2020) and DOM-Q-NET (Jia et al., 2019), both of which are fairly intricate algorithms. FreeDOM, for instance, requires a two-stage training process, and both algorithms utilize information from the entire DOM tree to represent a single element. We discovered that a surprisingly simple GCN, which solely uses information from local neighbourhoods around each node in the DOM tree, outperforms these SOTA methods. We further enhance the GCN by incorporating it into a Reversible GNN (RevGNN) architecture (Li et al., 2021). RevGNN improves both memory and parameter efficiency through reversible connections, group convolutions, and weight tying, enabling us to scale up to 20 layers and further boost the final nomination accuracy. These finding highlights the potential of employing GCNs as building blocks in more advanced web element nomination architectures in future research.

We subsequently investigate three methodologies aimed at further enhancing the web element nomination performance. Firstly, we explore whether the text present on the page is informative for nominating web elements. Specifically, we use a pre-trained *language model* to create embeddings of the text on the page, which we then add to the set of features. Our findings indicate that models incorporating text in general achieve higher average accuracy. Surprisingly, for certain tasks, the included text proves to be non-informative and, in fact, reduces performance. Secondly, applying LLMs to the entire HTML of realistic pages is unfeasible, partly due to limited context windows but mainly due to the high costs associated with processing a large number of tokens. To address this, we explore an approach where we significantly enhance the performance of the highest-performing GNN model with a post-training refinement step where an LLM is applied to a small subset of the highest ranked elements of each page. Finally, we address the absence of a standardized training methodology for element nomination[2]. To this end, we propose and evaluate a novel training approach that considerably improves nomination accuracy.

To summarize, our aim is to lower the entry barriers in this exciting field through the following contributions:

1. We introduce the *Klarna Product Page Dataset*[3], which consists of $51,701$ manually labeled product pages from $8,175$ e-commerce websites. The dataset is suitable for tasks such as element nomination,

---

[2]For comparison, see vertex nomination, as in Fishkind et al. (2015); Coppersmith (2014) This field typically involves very small numbers of graphs (in this context, webpages), rendering these methods unsuitable here.

[3]Available at https://github.com/klarna/product-page-dataset under the CC BY-NC-SA license.

and evaluation or fine-tuning of LLMs. We have also created a complementary dataset with the corresponding screenshots of the webpages for future CV applications.

2. We adapt and apply six algorithms from three of the most popular families of GNNs to a web element nomination task, namely recurrent GNNs (Tree Long Short-Term Memory networks), convolutional GNNs (GCNs) and attention based GNNs. We also investigate the performance of SOTA baselines for element nomination. In particular, we identify that *GCN-Mean*, a simple 2-layer GCN, outperforms the *second-best* model by an impressive margin of 9.2 percentage points measured in average nomination accuracy. We then enhance the scalability of this algorithm by employing RevGNN, which uses the GCN as a base model.

3. We explore a post-training nomination refinement step using an LLM. Specifically, we use a trained RevGNN algorithm to filter out all elements except the top ten for each class on every webpage in the test set. We then use GPT-4 to perform the final nomination based on the local HTML content of these filtered elements. Our results show that this refinement step substantially enhances nomination accuracy across all explored tasks, resulting in an average increase of 10.9 percentage points.

4. In element nomination, one would commonly train a model to classify elements and then evaluate the model based on its nomination performance. We refer to this approach as the *Basic Nomination Training Procedure*. To steer the classification objective towards the nomination objective, we introduce the novel *Challenge Nomination Training Procedure* (CNTP). Our results demonstrate that CNTP substantially improves nomination accuracy by approximately 5 percentage points for our best model.

5. We assess the impact that the text on the pages has on nomination accuracy by utilizing a *language model* to embed the text and integrate it as a feature. Our findings demonstrate that this considerably improves performance in specific nomination tasks, such as when identifying the Buy Button.

## 2 The Klarna Product Page Dataset

We collected the Klarna Product Page Dataset dataset over several months between 2018 and 2019. The dataset comprises 51,701 product pages from 8,175 distinct e-commerce merchants. It is conveniently divided into 80% training and 20% test sets, which ensures that pages from the same website do not appear in both the training and test sets. Table 1 presents an overview of the dataset statistics. Each page in our dataset is saved as an MHTML file, which includes all images and assets required to render the page. Furthermore, every page in the dataset has 5 labeled elements: 2 corresponding to action elements (*buy button* and *cart button*) and 3 to information elements (*product price*, *product name*, and *product image*). These labels have been manually annotated by human analysts. In our experiments, we also consider a 6th, more abstract label: the *subject node*, defined as the lowest common ancestor of all other labels. An example page screenshot

Table 1: An overview of the Klarna Product Page Dataset.

| market | language | # sites | # pages | median # nodes |
|--------|----------|---------|---------|----------------|
| DE | German | 2,941 | 16,765 | 1,391 |
| US | English | 1,794 | 11,003 | 1,394 |
| GB | English | 1,360 | 11,144 | 1,291 |
| FI | Finnish | 1,125 | 5,623 | 779 |
| AT | German | 899 | 1,316 | 1,499 |
| SE | Swedish | 619 | 4,866 | 1,450 |
| NO | Norwegian | 180 | 852 | 1,477 |
| NL | Dutch | 130 | 132 | 1,628 |
| Total | | 8,175 | 51,701 | 1,308 |

**Classification: Which Class does this Element belong to?**

**Nomination: Which Element is the Buy Button?**

Figure 3: **Difference between nomination and classification:** A DOM-tree representation of a webpage is depicted to the left. During the classification process, the GNN embedder takes a node and its context, here being its directly neighboring nodes, as input. The output from the embedder is then fed into a classification layer that produces classification scores. In element nomination, the model is applied to every node in the tree. For each label type, the resulting classification scores are ranked across all nodes, and the node receiving the highest rank is nominated for that specific label type. For example, here we see how the node receiving the highest ranking for the *Cart Button* label is nominated as the *Cart Button* element of the page.

from our dataset with marked labels is shown in Figure 2. Additionally, for convenience, we have created a complementary dataset of page screenshots, suitable for computer vision research.

## 3 Element nomination

Precisely identifying the correct element before executing any action in web automation is vital. In automation features, element nomination plays a crucial role as it solves the common subtask of identifying a single element on a webpage for a specific action. Since we often have only one opportunity to perform a task, incorrect actions, such as clicking the wrong button or filling in the wrong form, can degrade the user experience.

Unfortunately, there is no established training objective specifically for web element nomination (For further details on the complexities of formulating an objective for element nomination, see Appendix A.6) instead, the common practice is to train a model to classify elements and then assess its nomination performance. The distinction between classification and nomination is illustrated in Figure 3. During training, a GNN embeds a small subset of nodes from each page, and these embeddings are then passed to a classifier. After training, the model's performance is assessed based on its ability to nominate elements. In nomination, the task is to identify a specific element from a given class on each page. For instance, the cart button is nominated by applying the trained model to every node on the page to obtain classification scores. These scores are then ranked, and the node with the highest cart score is nominated as the cart element.

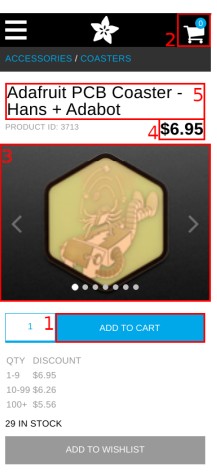

Figure 2: A sample webpage in our dataset with 748 nodes. The labeled elements are surrounded by red boxes. (1) is the *buy button*, (2) is the *cart button*, (3) is the *image*, (4) is the *price* and (5) is the product *name*.

To excel at the cart button nomination task, the classifier must consequently assign the highest cart score to the true cart element relative to all other elements on the page.

Element nomination is particularly challenging on realistic pages. A perfect classifier, which always assigns the correct class to every element, would also always nominate the correct element on every page. Hence, the classification objective serves as a proxy for training on the nomination task. However, the pages in our dataset consist of thousands of elements, and among these, there is a single labeled element for each class. For instance, a classifier with 99% accuracy would be expected to incorrectly classify 10 elements as 'add-to-cart' buttons on a page with 1,000 elements, even though only one true 'add-to-cart' button exists. Consequently, in Section 6.5, we propose an effective, novel training method that improves the element nomination performance by steering the classification training objective closer towards the nomination objective.

## 4    Related work

In the literature, there is an unfortunate scarcity of datasets suitable for web element nomination. In this section, we walk through and compare existing datasets to the Klarna Product Page Dataset.

The minimalistic World-of-bits environment (Shi et al., 2017b), known as Mini-WoB, is useful for task-solving on the Web. However, the pages in Mini-WoB are simple compared to sites found on the Internet. A page in Mini-WoB might, for instance, consist of just a single form and a submit button. In contrast, the Klarna Product Page Dataset comprises manually annotated clones of real webpages, with the median page featuring 1,308 elements and thousands of lines of code. The recently proposed WebShop (Yao et al., 2022) is another simulated interactive environment where agents can learn to navigate an e-commerce store and make purchases based on text instructions. WebShop includes 1.18 million products and 12,087 text instructions. While it effectively emulates an e-commerce store, its primary limitation is that it comprises pages from a single website.

There are datasets related to ours that are useful for web content extraction, perhaps the most well-known being the SWDE dataset (Hao et al., 2011). This dataset comprises 124,291 real webpages, each with $3-5$ labeled attributes, from 80 different websites across 8 verticals, such as online bookstores and camera retailers. Previous studies on SWDE (Lin et al., 2020; Zhou et al., 2021) have observed that test accuracy improves with the number of sites included in the training set for each vertical. However, the SWDE dataset is limited by having only 10 sites per vertical, thereby capping the potential gains. In comparison, our dataset consists of 8,175 diverse e-commerce websites, and thus provides a much more comprehensive representation of a vertical. Furthermore, Klarna Product Page Dataset encompasses labels that present a wider range of learning challenges. Whereas the SWDE dataset solely contains labeled leaf nodes, which are often identifiable based on local information, our dataset includes labels where local information alone is insufficient for accurate identification. Elements like the *price* and *name*, for instance, are primarily defined by local information, making their identification similar to standard web data extraction tasks. Conversely, correctly identifying the *buy* button often requires both local and contextual information. Lastly, the *subject node* has no meaning on its own and can only be represented based on its context. Lastly, unlike the SWDE dataset, which contains only English pages, our dataset includes pages with text written in multiple languages.

A recent large-scale webpage dataset that signifies a substantial advancement over previous works is Mind2Web (Deng et al., 2023). Mind2Web encompasses 137 websites across 31 diverse domains, such as Music, Airlines, Housing, and Social Media. Consequently, it offers a valuable opportunity to assess generalizability across various tasks and domains. Our dataset complements Mind2Web in this aspect, as Klarna Product Page Dataset enables the assessment of generalizability for the same task across a large number of websites.

Overall, we firmly believe that the diverse range of webpage structures and labels in our dataset marks a significant advancement over current SOTA datasets for web element nomination. However, we wish to emphasize that having access to a variety of datasets, each with different challenges and contexts, is invaluable. Such diversity is crucial for advancing toward truly autonomous and generalizable web agents.

## 5 Embedders for DOM Elements

We now describe the GNN embedders that we train together with a classification layer (see Fig 3) on our dataset. The embedders represent three GNN families, specifically recurrent, convolutional and attention-based networks. We now detail and then evaluate these embedders in the simulation section.

**GCN-Mean**. we first consider a multi-layer GCN model inspired by GraphSage (Hamilton et al., 2017) and PinSage (Ying et al., 2018). Each convolutional layer $\ell$, computes the representation of the current node, $v$, as $\mathbf{z}_v^{(\ell)}$ based on the local encoding and an average encoding $\mathbf{h}^{(\ell)}$ of its immediate neigborhood $\mathcal{N}(v)$. The representation is obtained according to the following:

$$
\begin{aligned}
\mathbf{h}_v^{(\ell)} &= \text{AVG}\left(\{\text{ReLU}(\mathbf{V}_l \mathbf{z}_u^{(\ell-1)} + \mathbf{b}_l),\ u \in \mathcal{N}(v)\}\right), \\
\mathbf{k}_v^{(\ell)} &= \text{ReLU}\left(\mathbf{W}_l \cdot \text{CONCAT}(\mathbf{z}_v^{(\ell-1)}, \mathbf{h}_v^{(\ell)}) + \mathbf{w}_\ell\right), \\
\mathbf{z}_v^{(\ell)} &= \mathbf{k}_v^{(\ell)}/\left\|\mathbf{k}_v^{(\ell)}\right\|_2,
\end{aligned}
\tag{1}
$$

where $\mathbf{V}_l$, $\mathbf{W}_l$, $\mathbf{b}_l$ and $\mathbf{w}_l$ are trainable weight matrices for convolution layer $\ell$ and $\mathbf{z}_i^{(0)} = \mathbf{x}_i^{(0)}$ for all $i$. We use the same dimension for the encodings $\mathbf{h}_v^{(\ell)}$ and $\mathbf{k}_v^{(\ell)}$. Encoding size, representation size, and total number of layers are tunable hyperparameters. By stacking $K$ convolution layers, information can be propagated from $K$-hop neighborhoods.

**TransformerEncoder**. This model consists of a single-layer multi-headed attention encoder stack (Vaswani et al., 2017) fed with a sequence consisting of the features $\mathbf{x}_v$ of the local node, $v$, stacked with the features $\mathbf{x}_u$ of its neighbors $u \in \mathcal{N}(v)$. The embedding $\mathbf{z}_v$ of node $v$ is then extracted as the first element (here, at index 0) of the sequence $\mathbf{H}_v$ computed by the transformer encoder stack:

$$
\mathbf{H}_v = \text{TransformerEncoder}\left(\text{STACK}\left(\mathbf{x}_v,\ \{\mathbf{x}_u,\ u \in \mathcal{N}(v)\}\right)\right), \quad \mathbf{z}_v = \mathbf{H}_v[0].
$$

In our implementation, the number of heads in the encoder and the embedding size are tunable hyperparameters.

We excluded positional encodings from this model, as they negatively impacted the transformer's performance. We explored several types of positional encodings, initially formulating them by traversing the DOM tree using both depth-first and breadth-first search orders. Additionally, we attempted, albeit unsuccessfully, to train a general, powerful, and scalable graph transformer (GraphGPS) (Rampášek et al., 2023) using Laplacian eigenvector positional encodings.

The ineffectiveness of positional encodings in element nomination might be due to the considerable variation in DOM tree structures across different websites. In our dataset, the number of nodes varies from a few hundred to approximately 20,000. However, a more consistent representation of page structure could be explored in future work by considering the positions of node elements in their corresponding rendered screenshots. For example, a screenshot of a page on an iPhone X device will have a consistent width across websites, and the position of elements in the screenshot tends to remain stable (e.g., the "go to cart" button is typically located in the upper right corner). This stability would make the position in the screenshot a more consistent and meaningful positional representation compared to the position in the DOM tree.

**Tree-LSTMs**. We consider four tree-based LSTMs. They all revolve around a standard LSTM cell (Hochreiter & Schmidhuber, 1997) and are characterized by how the input sequence is constructed from the DOM tree.

In the top-down model (LSTM-TD), the embedding $\mathbf{z}_v^D$ of node $v$ is the last element of the LSTM encoding of the path from the root to node $v$: each unit receives hidden and cell states from its parent, according to

$$
\mathbf{h}_v^D, \mathbf{c}_v^D = \text{LSTM}(\mathbf{x}_v, \mathbf{h}_{\text{Parent}(v)}^D, \mathbf{c}_{\text{Parent}(v)}^D), \quad \mathbf{z}_v^D = \mathbf{h}_v^D.
$$

where $\text{LSTM}(x, h, c)$ denotes a standard LSTM cell with input $x$, hidden state $h$ and cell state $c$. Cell and hidden states were initialized as zero for the root node.

In the bottom-up model (LSTM-BU), we consider the child-sum tree-LSTM algorithm (Tai et al., 2015). Here, the sequence is defined recursively from the leaves to each node $v$. At each node, the hidden state is the sum of the hidden states of the children, and the cell state is computed based on the cell states of the children, with one forget gate per child state.

$$\mathbf{h}_v^U, \mathbf{c}_v^U = \text{ChildSumLSTM}\big(\mathbf{x}_v, \{\mathbf{h}_u^U, u \in \mathcal{C}(u)\}, \{\mathbf{c}_u^U, u \in \mathcal{C}(u)\}\big), \quad \mathbf{z}_v^U = \mathbf{h}_v^U.$$

To account for more context when computing the element embeddings, we combined the bottom-up and top-down approaches into a bidirectional model (LSTM-Bi) where the embeddings are the concatenation of the two previous models' embeddings, $\mathbf{z}_v^B = \text{CONCAT}(\mathbf{z}_v^D, \mathbf{z}_v^U)$. In this case, the weights in the top-down and the bottom-up components are trained simultaneously.

Finally, we considered a global bidirectional model (LSTM-BiE) that computes node representations based on the whole DOM-tree (Cook, 2019). This method has three steps. First, a bottom-up representation $\mathbf{z}_v^U$ is calculated for each node $v$. Second, this representation is then used as the input to a top-Down architecture that gives a representation $\mathbf{z}_v^E$ for each node $v$:

$$\mathbf{h}_v^E, \mathbf{c}_v^E = \text{LSTM}(\mathbf{z}_v^U, \mathbf{h}_{\text{Parent}(v)}^E, \mathbf{c}_{\text{Parent}(v)}^E), \qquad \mathbf{z}_v^E = \mathbf{h}_v^E.$$

Third, the two representations are concatenated into a final representation for the node $\mathbf{z}_v^{BiE} = \text{CONCAT}(\mathbf{z}_v^E, \mathbf{z}_v^U)$. Also, in this case, the weights in both components are trained simultaneously.

**GCN-GRU**. This model is inspired by the local embedding module in the DOM-Q-Net algorithm (Jia et al., 2019). Here, the encoding of node $v$ is computed by feeding a Gated Recurrent Unit (GRU) with the local features of the node and an average encoding of the neighborhood of the node; then, the embedding is computed with a transformation of the local encoding $\mathbf{h}_v$ and the input features:

$$\mathbf{h}_v = \text{GRU}\big(\mathbf{x}_v, \text{AVG}(\{\mathbf{V}\mathbf{x}_u + \mathbf{b}, u \in \mathcal{N}(u)\})\big),$$
$$\mathbf{z}_v = \mathbf{W} \cdot \text{CONCAT}(\mathbf{x}_v, \mathbf{h}_v) + \mathbf{w},$$

where $\mathbf{V}$, $\mathbf{b}$, $\mathbf{W}$, and $\mathbf{w}$, are trained together with the parameters of the GRU.

**FreeDOM**. We adapt the FreeDOM architecture (Lin et al., 2020). This is a two-stage method where a neural network first computes node representations based on local and contextual information (using both text and HTML), and then uses representation distances and semantic relatedness between node pairs in the second stage. In the original implementation, (Lin et al., 2020) perform a filtering step where templates common to multiple pages are identified and used to reduce the number of nodes that the model considers. On our dataset, this is not possible since the pages generally do not come from the same set of websites. That being said, FreeDOM may be at a disadvantage compared to the other models because it does not perfectly suit our application or dataset. However, the authors set out to learn site-invariant feature representations and found that FreeDOM could generalize well to unseen websites.

## 6 Element Nomination Evaluation

As a starting point for research on the Klarna Product Page Dataset, we now compare various GNNs on the nomination task. To that effect, we first describe the common experimental setup in Section 6.1 and then present the results of this empirical evaluation in the following sections. These results first include the nomination accuracies from the Basic Training Procedure in Section 6.2. Next, we use the best-performing model from the initial evaluation to filter out most elements from the webpages and further refine our nomination results by applying GPT-4 to the HTML content of the filtered elements in Section 6.3. In Section 6.4, we conduct a sensitivity analysis on the best-performing GNN to investigate whether it suffers from over-smoothing and over-squashing. Lastly, we demonstrate that the CNTP training scheme significantly boosts nomination performance in Section 6.5.

### 6.1 Experimental Setup

**Training Procedures.** During training, the models learn to classify elements using the DOM tree representation of a page. More specifically, a GNN embedder takes a node with some neighborhood as input, and

Table 2: The nomination accuracy of each model on each label without (left) and with text (right) added to the element features.

| Model | buy btn w/o | buy btn w/ | cart btn w/o | cart btn w/ | main img w/o | main img w/ | name w/o | name w/ | price w/o | price w/ | subj w/o | subj w/ | avg w/o | avg w/ |
|---|---|---|---|---|---|---|---|---|---|---|---|---|---|---|
| LSTM-Bi* | .72 | - | .57 | - | .47 | - | .69 | - | .48 | - | .66 | - | .60 | - |
| LSTM-BiE* | .73 | - | .60 | - | .50 | - | .72 | - | .55 | - | .70 | - | .63 | - |
| GCN-GRU | .77(.01) | .83(.003) | .57(.005) | .62(.003) | .41(.006) | .38(.004) | .69(.001) | .79(.003) | .50(.02) | .60(.003) | .45(.003) | .53(.003) | .56 | .63 |
| FreeDOM** | - | .29(.09) | - | .056(.04) | - | .043(.02) | - | .57(.07) | - | .23(.03) | - | .35(.02) | - | .26 |
| FreeDOM-ext** | - | .84(.01) | - | .59(.007) | - | .31(.01) | - | .84(.01) | - | .72(.02) | - | .46(.02) | - | .63 |
| LSTM-TD | .71(.02) | .85(.04) | .50(.02) | .51(.02) | .42(.02) | .46(.02) | .70(.01) | .76(.00) | .52(.004) | .58(.03) | .59(.02) | .34(.01) | .57 | .58 |
| LSTM-BU | .73(.06) | .89(.03) | .57(.03) | .65(.02) | .45(.03) | .44(.02) | .69(.02) | .74(.01) | .46(.06) | .53(.02) | .66(.02) | .56(.02) | .59 | .64 |
| TE | .69(.02) | .81(.01) | .43(.03) | .46(.04) | .25(.06) | .34(.02) | .73(.03) | .79(.01) | .60(.03) | .62(.09) | .55(.03) | .62(.09) | .54 | .61 |
| FCN | .79(.01) | .86(.01) | .53(.004) | .62(.009) | .32(.03) | .27(.003) | .70(.01) | .85(.01) | .58(.09) | .61(.01) | .43(.04) | .39(.01) | .56 | .60 |
| GCN-Mean | .78(.04) | .91(.004) | .62(.005) | .67(.03) | .61(.02) | .48(.01) | .78(.004) | .81(.002) | .66(.02) | .66(.004) | .84(.01) | .86(.002) | .71 | .73 |
| REVGNN - 20 layer | **.81** | - | **.65** | - | **.62** | - | **.79** | - | **.66** | - | **.89** | - | **.74** | - |

its output is then classified by a single-layer fully connected neural network (see Fig 3 and Section 3). The classification gives rise to a cross-entropy training objective that is minimized using Adam. Furthermore, due to the large class imbalance between the six labeled nodes and the potentially thousands of unlabeled nodes on a page, we cannot train on all nodes of a page. In the *Basic Nomination Training Procedure*, presented in Section 6.2, every model instead classifies five labeled and ten randomly sampled unlabeled DOM elements from each page. In the *Challenge Nomination Training Procedure*, presented in Section 6.5, we conduct an additional experiment where we train on labeled nodes and unlabeled negative examples that the models confidently misclassified. Information regarding hyper-parameters can be found in Section A.1.

**Embedders.** We use embedders from three important GNN families, specifically recurrent, convolutional, and attention-based networks. Specifically, we consider LSTM-based embedders (LSTM-TD, LSTM-BU, LSTM-Bi, LSTM-BiE), the mean-pooled GCN model (GCN-Mean), the GRU-based GCN inspired by DOM-Q-Net (GCN-GRU), a 20 layer and 2 group RevGNN and the transformer-encoder (TE).[4] We also include a 20-layer, 2-group RevGNN model, which uses the GCN-Mean model as its base GNN. For comparison, we also present the results of the SOTA FreeDOM architecture, which we train on two different sets of features. First, we use the original features defined in (Lin et al., 2020), and then we specify an extended version (FreeDOM-ext) which, in addition, uses the style features employed by our other models (see the next paragraph). We also implement a 2-layer *Fully Connected Network (FCN)* that only looks at the features of a local node without considering any neighbors, thus acting as a context-oblivious baseline. After training, we evaluate how accurately the GNNs can nominate the six labels described in Section 2.

**Features.** For each node's *local features*, we define a set of *style features* which consists of the bounding box of the elements as rendered on the page, font weight, font size, number of images contained within the subtree, visibility, and HTML tag. For the FreeDOM-specific, pre-trained, NLP features described in (Lin et al., 2020), we use the Spacy[5](Honnibal et al., 2020) library. To also gauge the impact of textual features on the other models, we perform an experiment where we create an additional feature by embedding the text of a web element using the *Universal Sentence Encoder*[6] (Cer et al., 2018). FreeDOM's implementation stays the same in this experiment as it already uses text.

## 6.2 Basic Training Procedure

**Setup.** In this comparison we trained the models on a subset of the Klarna Product Page Dataset containing 10,000 pages from English markets. We did not train the models on the entire dataset because the LSTM models are very computationally expensive. It also offers a more level playing field for the SOTA FreeDOM architecture, since it relies on page text and was originally applied to English websites.

---

[4]We would also like to address the absence of the very popular Graph Attention Network (GAT) from our evaluation. Despite investing significant effort in training various versions of GAT on our dataset, its performance was consistently inferior to the other models we assessed. Consequently, we decided to exclude GAT from our evaluation.

[5]v2.3.5 *en_core_web_sm* https://spacy.io/models/en (MIT License)

[6]Version 4, https://tfhub.dev/google/universal-sentence-encoder/4 (Apache-2.0 License)

**Evaluation.** The trained models are evaluated based on their nomination performance. As described in Section 3, with the exception of FreeDOM, we compute the nomination metric by classifying all nodes on every webpage in the test set. These scores are used to rank the nodes for each of the classes: *price*, *name*, *image*, *buy button*, *cart button*, and *subject node*. The node with the highest score for each class is nominated for that class. A nomination is considered correct if the nominated element is the single element with the ground truth label. For FreeDOM, we followed the protocol described in (Lin et al., 2020). The evaluation is performed on 6, 706 test pages from merchants not present in the training set.

### 6.2.1 Results

**Nomination Accuracy.** Table 2 contains the nomination results. Note that here precision and recall both equal accuracy as the methods make exactly one prediction on each page, and there is one correct element per class. From the results with only style features (no text features), we first notice that GCN-Mean achieves the highest average accuracy by a wide margin (6.8 p.p.). Despite its relative simplicity, GCN-Mean consistently scores among the best for all tasks explored here. Since GCN-Mean uses k-hop neighborhoods (see Section 5), this suggests that a large proportion of the context relevant for element nomination is concentrated in an element's vicinity. The RevGNN algorithm successfully builds upon the GCN-Mean algorithm, leading to further improvements in the final nomination accuracy. Among the runner-ups, LSTM-BiE stands out, as it shows similar performance to FreeDOM-ext (using text), while GCN-GRU, LSTM-TD, LSTM-Bi and LSTM-BU all show similar average performance. It is worth noting that FreeDOM requires two separate training stages and that the LSTMs require substantially more compute than GCN-Mean, GCN-GRU and FreeDOM.

**Classification Accuracy.** The classification accuracy of the models considered here are presented in Section A.5. Despite all models achieving very high and similar classification scores, their nomination performance differs substantially. This illustrates the discrepancy between the classification training objective and the nomination evaluation task. We address this discrepancy using CNTP, which we present further below.

**The Impact of Contextual Information on Nomination.** We found that more context helps performance on certain tasks. As an indicator of the importance of context, we consider the performance of the FCN. Specifically, since the FCN only looks at a local node's features without including information from neighbors, the gap between the best model and the FCN gives us a lower bound on the gain we can achieve by including contextual information. With this contextual importance indicator, it is possible to identify tasks where contextual information is more important. For example, the *subject node* on its own does not contain much information but is instead defined by its relationship to other nodes in the tree; therefore, we need to look at its context to identify it. For other tasks, context is less important. For example, the *buy button* seemingly contains sufficient information to be correctly identified based on its local features. Finally, another interesting observation related to context is that LSTM-BiE outperforms LSTM-Bi on all tasks, indicating that processing all context from the page improves accuracy.

**The Impact of Text on Nomination.** When we add text features, GCN-Mean still has the highest average accuracy, this time by an even wider margin. Though LSTM-BiE was not trained with text due to computational constraints, its performance remains competitive with all other algorithms except GCN-Mean when trained on solely style features. This further solidifies our intuition that straightforward graphical algorithms such as GCN-Mean and LSTM-BiE should be explored further as building blocks in web element nomination algorithms. Furthermore, one important benefit of LSTM-BiE and GCN-Mean over FreeDOM, SimpDOM, and WebFormer is that they remain competitive without text features. This allows us to deploy a single model that can be used across webpages, regardless of the language used on the page.

Another important finding is that algorithms that try to exploit text perform worse when this information is irrelevant to the task. We can observe this degradation when GCN-Mean, GCN-GRU, and even FCN attempt to use text to nominate the image node. LSTM-TD performance drops when text is added to its feature set since it looks at the text in nodes above the *subject node*. The *subject nodes* are usually close to the root of the DOM tree; hence, it is unlikely that nodes in the considered sequence even contain text. This variety in performance suggests that we need varying degrees of contextual information to nominate

different label types. This also suggests that the Klarna Product Page Dataset presents a comprehensive set of challenges that not all algorithms can easily overcome.

### 6.3 Post Training LLM Refinement

**Setup.** The refinement step was initiated by filtering out all non-informative elements from the test set using the trained 20-layer RevGNN algorithm from Section 6.2. This filtering was performed by ranking the elements by their classification scores and retaining the ten highest-ranked elements. The raw HTML of these elements, combined with their bounding boxes, was then used to formulate a query where the task was to select the single labeled element from the set of elements. We then presented this query to GPT-4, which performed the final nomination by providing an answer to the query. This experiment was constrained to 500 randomly sampled English pages from the test set.

**Results.** From the results shown in Figure 4, it is evident that the LLM refinement step substantially enhances the final nominations, despite relying solely on local element content. It significantly boosts the nomination performance across all tasks, resulting in an average nomination accuracy increase of 10.9 percentage points. The improvement is particularly noticeable for the product image element, indicating that the raw HTML contains substantial information for this specific task. However, there is still room for improvement, which could potentially be achieved by incorporating screenshots of the rendered pages.

Figure 4: **Substantially Enhanced Nomination Accuracy by Combining RevGNN with GPT-4:** The performance of the RevGNN algorithm alone versus its performance when combined with a GPT-4-based refinement step. In the enhanced scenario, the RevGNN algorithm initially filters the dataset to retain only the top 10 elements with the highest classification scores on each page. Subsequently, GPT-4 undertakes the final nomination step, based on the local HTML content of these selected elements. This refinement step significantly boosts nomination accuracy across all tasks, resulting in an average increase of 10.9 percentage points.

### 6.4 Sensitivity Analysis

Simple message-passing GNNs may suffer from over-smoothing and over-squashing, where information flow between distant nodes becomes distorted (Topping et al., 2021). This distortion can negatively impact the model's performance in tasks that rely on long-range interactions, such as element nomination, where critical information may be located far from the root in large trees. To assess whether the GCN-Mean algorithm is prone to these issues, we conducted a sensitivity analysis on a subset of 8,000 pages, gradually increasing the number of layers, as shown in Table 3. The results indicate that while the GCN's performance initially improves with more layers, it begins to degrade after adding a third layer, suggesting the presence of over-smoothing and over-squashing effects.

One potential approach to mitigating these effects is graph rewiring, where the DOM-tree is modified to enhance information flow. A particularly promising method that could be explored in future work is Expander Graph Propagation (Deac et al., 2022), which augments the DOM-tree with expander graphs. These graphs, despite their sparsity, ensure efficient message passing throughout the entire structure, enabling information to propagate quickly and effectively. This approach could be well-suited for managing the complexities of large DOM-trees, where critical information may be dispersed across the tree.

### 6.5 Challenge Nomination Training Procedure - Bridging the *classification* and *nomination* objectives

We put forth CNTP, a novel two-stage training approach which is similar to the method in Ben-Baruch et al. (2022) and inspired by Positive-Unlabelled Learning (see Bekker & Davis (2020) for a survey). In CNTP, we

| No. Layers | 1 | 2 | 3 | 4 |
|---|---|---|---|---|
| Avg Nomination Accuracy | 0.649 | 0.679 | 0.666 | 0.631 |

Table 3: Sensitivity analysis results from varying the number of layers in the GCN-Mean Algorithm.

train on the classification objective, which means that we only need to train on a small subset of elements from each page in the training set. We then periodically rank the elements on the page based on their classification scores and add the confusing elements to the training set. We define a confusing element as an unlabeled element that inaccurately receives a very high score for a particular label. This procedure increases the likelihood that the true labeled elements appear at the top of the ranking during the nomination phase.

**Notation.** The parameters of CNTP are: $M$, the number of unlabeled elements to be randomly included in the training set from each page; $T$, the number of training epochs; and $K$, the number of additional *hard* elements identified by the model in the nomination task after $T$ epochs.

**CNTP.** The pseudo-code for the training data augmentation step is presented in Algorithm 1. A description of the procedure goes as follows: At the start of every epoch, we begin with an empty training set of elements $S_{train} = \varnothing$. For each webpage $P \triangleq (V, E)$, where $V$ is the set of elements in the training set $\mathcal{P}$, we uniformly sample a set $S_P = \{v_i \in V_{unlabelled} : i = 1, \ldots, M\}$ of $M$ unlabeled elements. We add $S_P$ to the classifier's training set, along with all $L$ labeled elements: $S_{train} = \bigcup_{P \in \mathcal{P}}(S_P \cup V_{labelled})$. We then train the model for one epoch. At epoch $T$, we evaluate the model on the nomination task across the entire set $\mathcal{P}$ and initialize the sets of *hard* training examples, $H_P = \varnothing$ for all $P \in \mathcal{P}$. Then, for all $P \in \mathcal{P}$, we add the top $K$ unlabeled elements in the proposed ranking for each of the $L$ labels to $H_P$. It is sufficient to perform this step once. For all future epochs, these elements are always considered additional training examples in the classification task: $S_{train} = \bigcup_{P \in \mathcal{P}}(H_P \cup S_P \cup V_{labelled})$. Thus, we train on $M + L \times K$ unlabeled elements and all the labeled ones. Note that, as opposed to Ben-Baruch et al. (2022), we attach the *negative* label to these nodes.

CNTP offers two main benefits: 1) it uses only a small fraction of the total elements in the training set, and 2) it bridges the gap between the nomination and classification tasks. The first benefit allows us to avoid the impractical computational cost of training to rank all elements, focusing only on the top candidates instead of those confidently identified as uninteresting. The second benefit is achieved by training on more confusing unlabeled elements—those likely to appear high in the nomination ranking. This approach aids the model in distinguishing between the ground truth elements and the uninteresting elements that receive high scores.

### 6.5.1 Results

Here we use the entire dataset with pages from all markets, with $M = 20$, $K = 5$, $T = 50$, and $L = 6$.

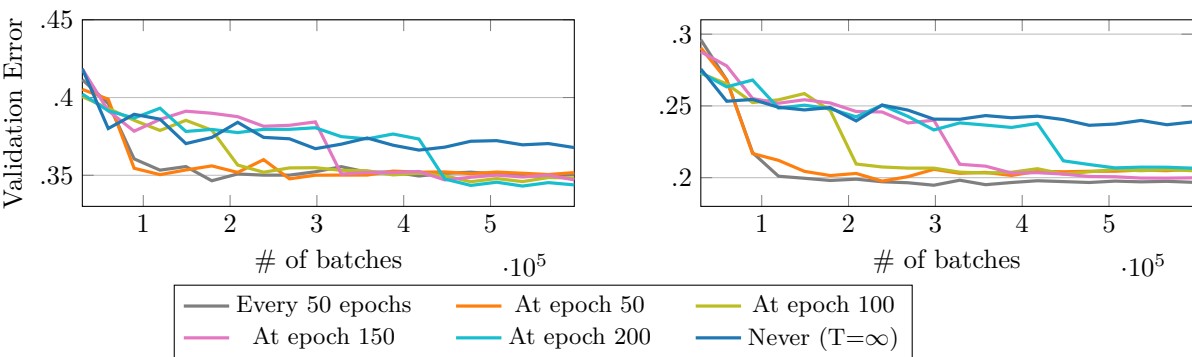

Figure 5: Effect on average validation error from performing the augmentation step at different times, consistently, and not at all for FCN (left) and GCN-Mean (right).

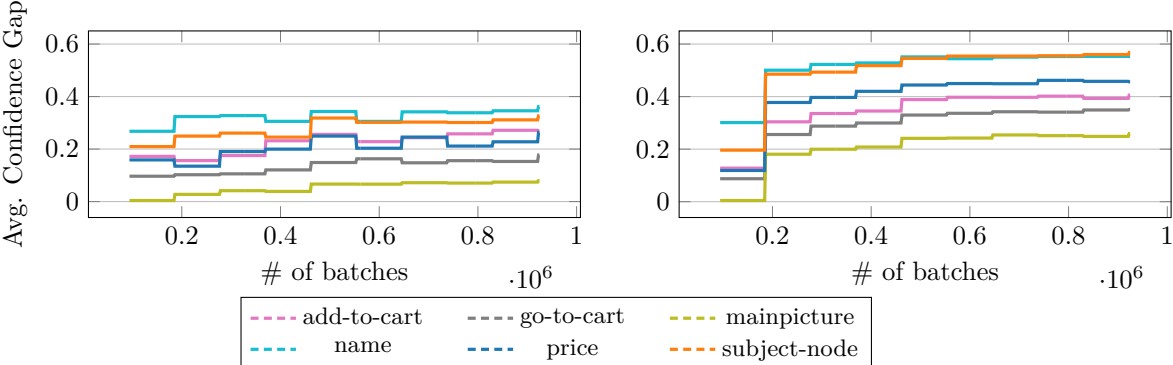

Figure 6: Effect on the average *confidence gap* from using *hard examples* for GCN-Mean. The left plot shows the average confidence gaps during training without augmentation. On the right, we show the corresponding confidence gaps when the hard examples are added to the training set.

**Effects on Nomination Accuracy.** Figure 5 highlights the effect of the timing of the *hard example* augmentation step on the average nomination accuracy on the validation set while training the FCN and GCN-Mean. We observe that the timing of the augmentation makes little difference on the final nomination accuracy. This suggests that there are a few consistently confusing elements that can be relatively easily identified using models trained on randomly selected unlabeled elements as negatives. Once these elements are in the training set, the models' nomination performance improves significantly. Indeed, both FCN and GCN-Mean achieve lower errors after the augmentation, by about 2 and 5 percentage points, respectively, with GCN-Mean reaching beyond 80

**Effects on the Quality of Rankings.** We evaluate whether CNTP brings us closer to training directly on the nomination objective. As a proxy fort his, we measure the *confidence gap*: the difference between the confidence assigned to the labeled element and the highest confidence assigned to any unlabeled element on the same page. We assume that the larger this difference is, the *better* the resulting ranking will be. Since either unlabeled elements become less likely to be mistaken for the true element (when the true element is ranked first) or the difference by which an unlabeled element wins in the ranking (when the true element is not ranked first) is lower, and hence the ground-truth labeled element appears higher in the ranking.

In Figure 6, we ask the models to perform the nomination task on the entire training set every 50 epochs and measure the average confidence gap per class. From the basic classification objective, we observe that the confidence gap increases on average, yet this behaviour is far from monotonic. In contrast, when we add *hard examples* during training, the rankings that the nomination objective is based on gets steadily better when we add *hard examples* during training according to our metric. This performance more closely resembles what we would expect when optimizing directly for the nomination objective. In this experiment, augmentation is performed every 50 epochs, though the same behavior is observed when this step is performed only once.

## 7 Limitations

One shortcoming of the Klarna Product Page Dataset is that far from all elements have a ground truth label; we only have one label per class per page, which makes our results slightly pessimistic. In reality, a click event on several elements within the *buy button* (e.g., the button itself and the node containing the button text) could generate the same result. Thus, there are several acceptable nominations in practice. The *single label per element type* issue reduces the number of positive training examples in the training set and increases the difficulty of the nomination task. While this results in a lower overall accuracy than what would be achieved in practice, it should not affect the comparison between methods. Furthermore, perfectly annotating all nodes of each element in a dataset of this size is very challenging and would require substantial web development knowledge. One approach to reduce the stringency of the evaluation would be to check the

content of nominated elements. However, this is not appropriate for action elements and requires resilient content extraction heuristics for all considered classes.

## 8 Conclusion

We introduce Klarna Product Page Dataset, a large-scale realistic dataset containing $51,701$ product pages from $8,175$ merchants across 8 markets, with labels that present varied challenges. To initiate research on our dataset, we explored the potential of GNNs for nominating webpage elements and how these methods could be combined with LLMs. Our experiments demonstrate the untapped potential of GNNs for web element nomination and illustrate how to effectively combine GNNs with LLMs for this purpose.

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
