# A    Appendix

## A.1    Hyper-Parameters

We ensured that all hyper-parameters were optimized for this setting. For FreeDOM, we used the hyper-parameters provided by the original paper since our setting is very similar. For the TreeLSTM-based algorithms, we used the results of the hyper-parameter tuning in Cook (2019). We then performed hyper-parameter tuning for GCN-Mean, GCN-GRU, FCN and the TransformerEncoder. We tuned our method and these other particular algorithms since we have adapted and applied them to a substantially different setting than what they were designed for. In the tuning procedure, we performed a grid-search over the values in HYPERPARAMETER_GRIDS folder[7] using the RAY TUNE (Liaw et al., 2018) library.

## A.2    Bridging Classification and Nomination

Algorithm 1 details the procedure we follow for augmenting the sample of web elements used in training on the classification objective for it to better resemble optimisation on the nomination task. In Figure 7, we can see the evolution of the average confidence assigned to the true labelled elements (continuous lines) versus the average maximal confidence assigned to any unlabelled element (dashed lines). We can clearly notice the distance between these quantities increasing dramatically after the augmentation step at epoch 50 and also, them continuing to diverge as training continues.

---

**Algorithm 1** Hard-Example Augmentation

---

1:  Inputs: $M, K, \epsilon, \mathcal{P}$ (the page dataset)
2:  $H_P := \varnothing, \forall P \in \mathcal{P}$
3:  nn_model := model_init()
4:  **for** $epoch = 1, 2, \ldots$ **do**
5:      $S_{train} := \varnothing$
6:      **for each** webpage $P \in \mathcal{P}$ **do**
7:          $P \triangleq (V, E), V \triangleq V_{labelled} \cup V_{unlabelled}$
8:          $S_P := \text{UniformRand}(V_{unlabelled}, \text{n}_{\text{samples}} = M)$
9:          **if** $epoch = K$ **then**
10:             **for** $l := 1, \ldots, L$ **do**
11:                 $preds = \{\text{nn\_model}(v)[l], \forall \text{elements } v \in V\})$
12:                 $H_P := H_P \cup \text{rank}(preds)[: K]$
13:             **end for**
14:         **end if**
15:         $S_{train} := S_{train} \cup S_P \cup V_{labelled} \cup H_P$
16:     **end for**
17:     Train nn_model for 1 epoch on dataset $S_{train}$.
18: **end for**

---

## A.3    Nomination Accuracy Results

In Figure 8 we present a graphical representation of the nomination accuracy results in Table 2. In this representation, in order to better visualise the impact of text features, the performance of an algorithm when using text features and not, are overlaid: full bars represents performance without text features and overlaid on top, in hashed/transparent bars is the performance when using text.

## A.4    Accuracy over Amount of Training Data

In this experiment, we aim to check how the nomination accuracy of the proposed methods increases as they are allowed to train on more data. Our goal is to verify whether any of the algorithms show signs

---

[7]The code, along with all the hyperparameters, is available at https://github.com/Anonymised/product-page-dataset.

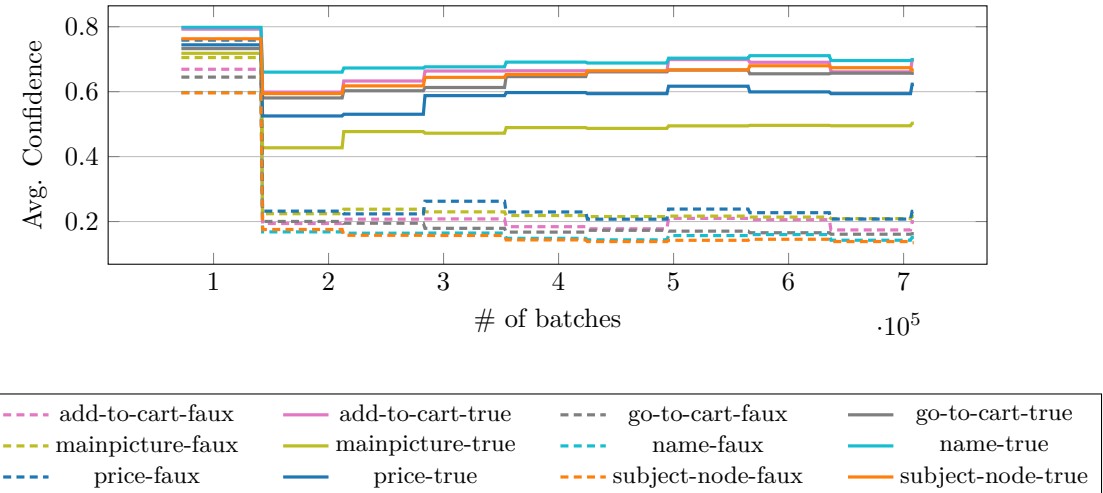

Figure 7: Average confidence assigned by GCN-mean to the true elements are displayed in solid lines, (with suffixes "-TRUE") and the average maximum of the confidence scores of all unlabelled elements on a page are in dashed lines ("-FAUX").

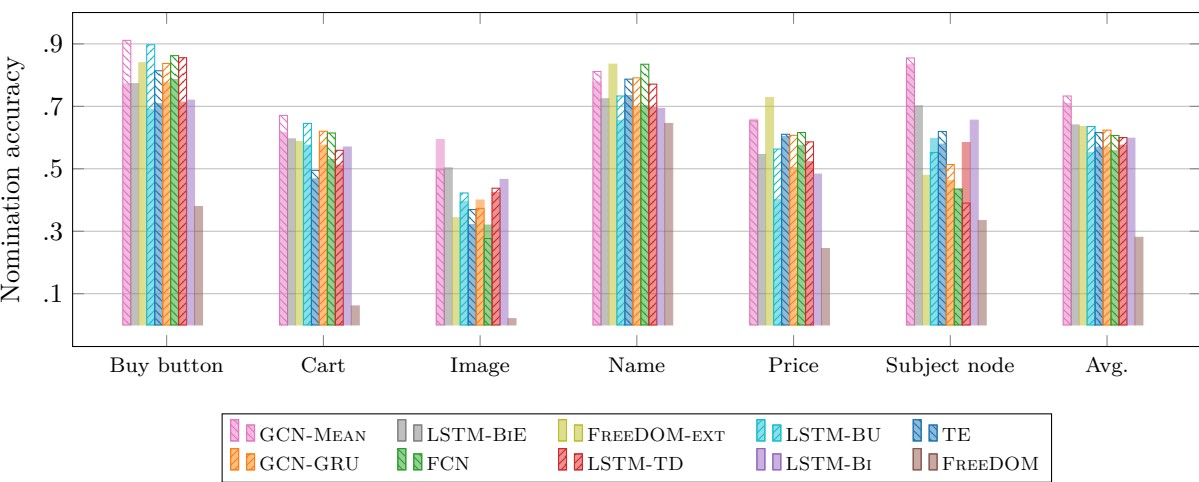

Figure 8: Nomination accuracy with (overlaid dashed bars) and without (solid bars) text included in the local element features.

of performing substantially better as the training set grows. If this were the case, it would suggest the further experiments should be run using a larger training set. We therefore trained the models on 5 subsets from the Klarna Product Page Dataset increasing in size by powers of 2, from 128 to 2048 DOM trees and an additional 6-th dataset containing the 10,000 webpages in the previous experiment. Each model was trained with fixed hyperparameters, all models use internal representations of size 150, for 50 epochs on every dataset, on only the local style features. The average predictive performance over the 6 labels of each model over the 6 datasets of increasing sizes is displayed in Figure 9. Here, again we see that, among the best performing algorithms, GCN-MEAN also exhibits the largest accuracy gain from adding more data. This raises our confidence that convolutional neural networks are worth considering in more detail for web element nomination tasks.

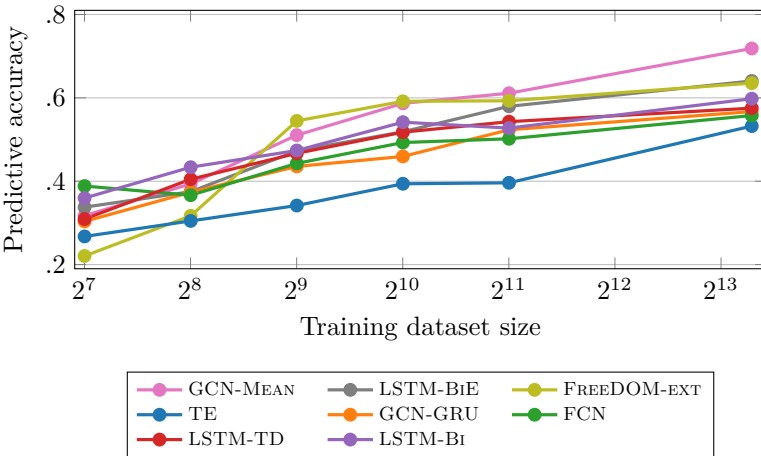

Figure 9: The nomination accuracy (averaged over classes, measured on the validation set) of each algorithm, when trained on the same training sets of increasing size.

## A.5 Classification Performance

We also provide additional precision and recall metrics corresponding to the performance of the trained models evaluated in Table 2. Table 4 details the classification performance of the models without text features. When using text features, the results are very similar hence we do not present them here to save space. These metrics show that all models achieve high classification performance, pointing towards them being sufficiently trained on this objective.

## A.6 Formulating an Element Nomination Objective

In this section, we discuss the complexity of formulating an objective for element nomination. At first glance, it might appear feasible to design an architecture where the final layer outputs one value per node and class, applying a softmax function across all nodes for each class, and then directly learns to nominate elements using a cross-entropy loss. However, in the following sections, we will outline several challenges and limitations associated with this approach.

To understand why this is not straightforward, assume that we have $L$ class labels and that we for now disregard the unlabeled class. Thus, if we have $N$ nodes on a page, we could use $L$ one-hot vectors of length $N$ to encode which nodes belong to each class on a page.

Table 4: Final classification accuracy of trained models (Precision/Recall, trained without text features)

|  | buy button | cart button | product image | name | price | subject node | avg. |
|---|---|---|---|---|---|---|---|
| LSTM-Bɪ | .91/.95 | **.92/.95** | **.89/.94** | .93/.89 | .85/.90 | .96/.91 | .95/.95 |
| LSTM-BɪE | **.97/.92** | .95/.89 | .92/.85 | .92/.91 | .90/.86 | **.96/.96** | .95/.95 |
| GCN-GRU | .95/.92 | .93/.86 | .90/.88 | .90/.90 | .89/.85 | .95/.93 | .94/.94 |
| Fʀᴇᴇᴅᴏᴍ | .80/.93 | .45/.88 | .50/.43 | .91/.88 | .86/.88 | .96/.33 | .84/.82 |
| Fʀᴇᴇᴅᴏᴍ-ᴇxᴛ | .95/.95 | .92/.88 | .89/.90 | **.97/.95** | **.96/.94** | .92/.91 | **.95/.95** |
| FCN | .96/.91 | .94/.81 | .91/.80 | .89/.91 | .83/.88 | .93/.94 | .93/.93 |
| LSTM-TD | .94/.93 | .93/.86 | .90/.83 | .92/.91 | .90/.85 | .94/.90 | .94/.94 |
| TE | .91/.94 | .90/.94 | .88/.91 | .92/.90 | .84/.93 | .95/.91 | .95/.95 |
| GCN-Mᴇᴀɴ | .97/.90 | .96/.80 | .94/.85 | .93/.92 | .93/.84 | .97/.91 | .94/.94 |

Imagine that we map the hidden representations of the $N$ nodes on the page into an $N \times L$ matrix using a function $f_\theta(\cdot)$ and subsequently apply softmax along the nodes for all $L$ classes. Then we could formulate a cross-entropy loss function as:

$$\sum_{l=1}^{L} \sum_{n=1}^{N} p(x_{n,l}) \log q(x_{n,l}). \tag{2}$$

After incorporating the one-hot encoded labels, this double sum just becomes a sum over $L$ terms, one for each class.

The reason why this objective does not work is that it does not instruct $f_\theta(\cdot)$ how to properly handle unlabeled nodes. When $f_\theta(\cdot)$ maps a labeled node into the $L$ class probabilities, it simply assigns a high amount of mass to the $l$-th correct class. However, for an unlabeled node, it should not learn to assign any mass to any of the $L$ classes, yet because it is a mapping to these $L$ classes it is forced to distribute its mass among these classes somehow.

A natural suggestion for handling this issue is to incorporate the unlabeled class such that $f_\theta(\cdot)$ maps the node embeddings into an $N \times (L+1)$ matrix. Now it remains to decide how to represent which nodes belong to which class. For the labeled nodes, we keep the $L$ one-hot vectors of length $N$ to encode which nodes belong to each class. However, it is now unclear how to encode the unlabeled nodes. One approach could be to use a vector of $N - L$ ones and $L$ zeros, where $N - L$ corresponds to the number of unlabeled nodes on the page.

The issue with this approach is related to class imbalance. After incorporating the labels into the equation above, we now have $L + (N - 1)$ terms, where $L$ terms incentivize the model to identify the $L$ labeled nodes correctly, while $(N - L)$ terms (where $N$ can be up to $20,000$) incentivize the model to learn to recognize unlabeled nodes correctly. With this very large imbalance, the model might as well predict that everything is unlabeled.

One suggestion to mitigate the class imbalance could be to uniformly weight the terms in the loss which correspond to the unlabeled nodes uniformly, resulting in the following cross-entropy formulation:

$$\sum_{l=1}^{L} \sum_{n=1}^{N} p(x_{n,l}) \log q(x_{n,l}) + \frac{1}{(N-L)} \sum_{n=1}^{N} p(x_{n,(L+1)}) \log q(x_{n,(L+1)}). \tag{3}$$

The issue with this approach is that the model is now less incentivized to correctly identify every single unlabeled node. The implication is that when it encounters difficult negative examples, it is less punished for incorrectly assigning more mass to one of the $L$ labeled classes instead of the unlabeled class. Not learning to handle hard negative examples can significantly degrade the final nomination performance, as it just takes a single confusing element to ruin the nomination.

### A.7 Model Failures

To gain some intuition about the limitations of our models, we include visualizations of instances where the 20-layer RevGNN algorithm fails, as shown in Figure 10.

### A.8 Klarna Product Page Dataset - Additional Statistics

The histograms in Figure 11 show an overview of the complexity that is expected from the pages in our dataset and aim to complement Table 1. The top two plots, displaying the distribution of the size of trees in both heights and the total number of nodes, give a feel for the length of sequences on which the TreeLSTM methods operate and the effect of the number convolutional layers and sizes of the filters. The number of images gauge the amount of noise to be expected when dealing with page screenshots.

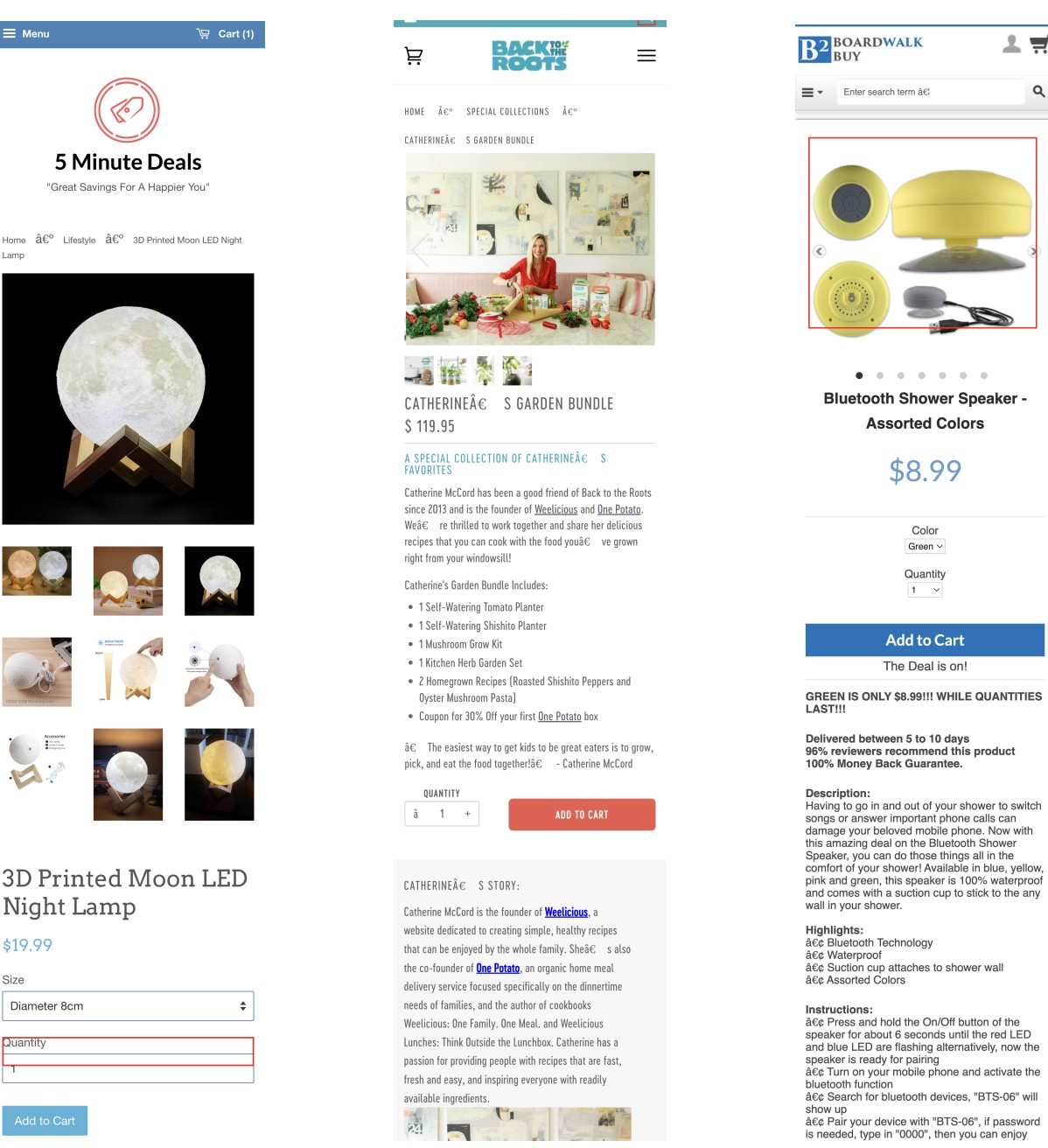

Figure 10: Three page screenshots where the model's nominations are inaccurate, with incorrectly identified elements outlined by bounding boxes. In the screenshot on the far left, the model is tasked with identifying the "Buy" button but mistakenly highlights an element located below the product image and price in the center. While the "Buy" button can sometimes be found in this area, it is not accurate in this instance. In the middle image, the model incorrectly identifies a search button in the upper right corner as the "Go to Cart" button, which, although commonly located in this area, is not correct in this context. In the final screenshot on the right, the model inaccurately identifies a div element placed over the product image as the product image itself. These examples illustrate a limitation of relying solely on the DOM tree, as developer design choices are not always consistent or tidy.

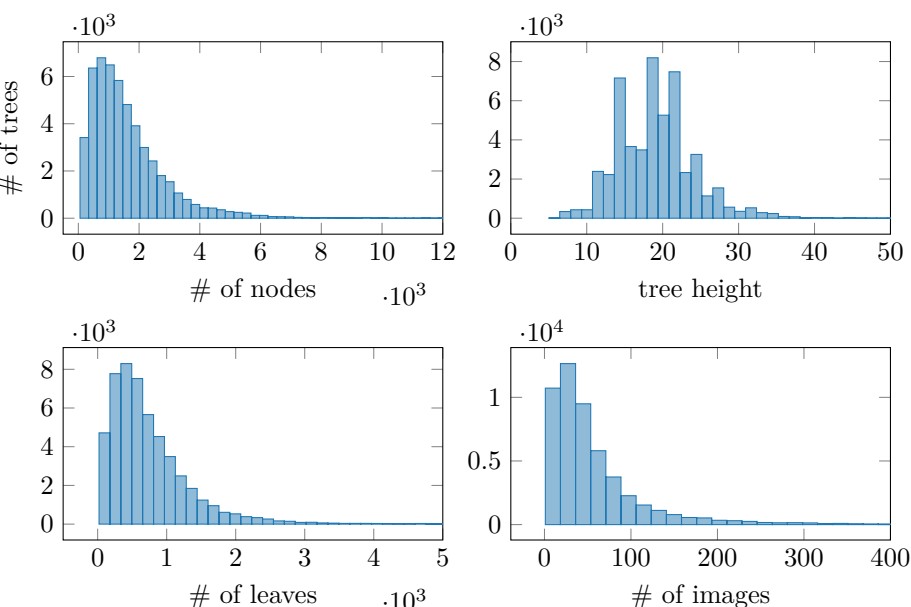

Figure 11: The histogram in the top left corner depicts the number of nodes per tree. To its right, the tree heights are displayed. A tree's height is the maximum possible path length from the root of the tree. In the bottom left corner, we give the number of leaf nodes per webpage, and to its right, we display the number of images (png, jpg, gif) per webpage.