# OpenReview forum: "The Klarna Product Page Dataset: Web Element Nomination with Graph Neural Networks and Large Language Models"
_TMLR — Accepted by TMLR_

### Review · Reviewer_grL9 · 2024-07-28

**Summary Of Contributions:**

The paper introduces a new dataset that consists of a collection of more than 50,000 manually labeled product pages from 8,175 e-commerce websites. The websites cover 8 different markets and 5 languages, and there are 5 labelled elements per page. To illustrate a use-case for the dataset, the paper presents benchmark results for the web element nomination task, comparing GNN methods with existing techniques tailored to the specific problem. The paper provides results showing that using a GCN to select a set of candidates and then asking an LLM to select the single answer leads to significantly improved performance. Finally, the paper presents a modified training approach that is more closely aligned to the nomination task than the standard classification-proxy technique. The paper shows that this improves performance and provides experimental results that explore the impact of including the modifications at different stages of the learning process.

**Audience:**

Yes

**Claims And Evidence:**

Yes

**Requested Changes:**

R1.	Please add other GNNs to the experimental results or explain why it is sufficient to include only the GCN. It would be helpful to see results for GAT and GIN. Beyond this, one or two of the more recent methods should be included. RevGNN (Li et al. 2021) and DRGCN (Zhang et al. 2023) are often amon the best performing. GraphGPS or GRIT could serve as suitable graph transformer models. [CRITICAL]

R2.	Please discuss or explain why it would not make sense to apply a heterogeneous GNN (available in PyTorch Geometric, for example). I don’t have a full appreciation for the dataset, but it would appear that there is a distinction between parent and children relationships and that this should be taken into account by the GNN. [STRENGTHEN]

R3.	Please provide a clearer explanation of why the nomination problem is approached as a classification task. This may be obvious to someone who works on this topic, but the paper doesn’t convey it currently. A brief discussion of the problem, with more details about the constraints that motivate the classification proxy, could be included in an appendix. From the current text, it is not clear to me why one would not construct an architecture with an output layer that takes in the embeddings of all nodes and applies a softmax (effectively assigning a probability to each node that it is the unique element of interest). Then one could perform learning using a cross-entropy loss. A deep-set type architecture could ensure equivariance to the ordering of the nodes. [STRENGTHEN]

R4.	Please provide a clearer explanation of why positional encoding is not feasible in this setting. Many of the graph transformers have effective positional encoding strategies. These can be relative, like shortest-path distance or relative random walk probabilities, or absolute, potentially based on eigenvalues. Why are these not usable for this particular dataset? In the tree setting, the depth would also seem to be a natural encoding. [STRENGTHEN]

R5.	Please provide a better presentation of the variability of the results. I would encourage bootstrap confidence intervals, which are very straightforward to generate with scipy.stats.bootstrap, without requiring multiple experimental trials. Beyond this, it would be helpful to see the results of hypothesis testing for significance of the results. The Wilcoxon paired signed rank test can probably be used (or Kruskal-Wallis with Dunn as a post-hoc). If normality assumptions are satisfied, then the pairwise t-test is an alternative. [CRITICAL]

R6.	Please provide suitable sensitivity studies. The paper hypothesizes that a GCN with few layers (2 or 3?) is the most effective, indicating that local neighbours are the most relevant. Please report the performance of a GCN with varying number of layers to quantify and support this observation. The GCN, and message-passing GNNs in general, are prone to over-smoothing and over-squashing, and struggle to incorporate pertinent information from nodes that are multiple hops away. This is where it would be helpful to see the performance of other, more recent GNNs that have explicit approaches to avoid this (e.g,, directly incorporating second-hop neighbour information without requiring that it is first passed through the first-hop neighbours). [CRITICAL]

**Strengths And Weaknesses:**

Strengths

S1. The paper introduces an impressive dataset that should be of interest and value for researchers working on multiple problems.

S2. The paper provides an example use-case for the dataset, providing benchmark results for the web element nomination task.

S3. Results are presented that illustrate that refinement of results using an LLM can significantly improve performance.

S4. The paper describes a modified training approach that is more closely aligned with the nomination task.

Weaknesses

W1. The benchmarking does not include many GNNs. Basic GNNs like GAT and GIN, which are widely used, are not included. There is no experimentation with state-of-the-art GNNs such as graph transformers. There is no exploration of heterogeneous GNNs.

W2. The benchmarking does not include any sensitivity analysis. For example, the paper does not report on the impact of the number of layers in the GCN. Results are only reported for the mean aggregation strategy.

W3. There is no reporting of the variability of performance. The paper should report confidence intervals, or at least standard deviations. There is no hypothesis testing to establish that the observed performance differentials are statistically significant.

---

> ### Author Response · Authors · 2024-08-26
>
> # R1
> We have included the requested results for the RevGNN models using the following [implementation](https://github.com/pyg-team/pytorch_geometric/blob/master/examples/rev_gnn.py). For the RevGNN model, we used 2 groups, 20 layers, and the GCN-Mean algorithm as the base GNN ([see documentation](https://pytorch-geometric.readthedocs.io/en/latest/generated/torch_geometric.nn.models.GroupAddRev.html)). We also trained a 100-layer RevGNN model, but observed a degradation in performance. A comparison between the 20-layer RevGNN and the GCN-Mean algorithm is presented in the table below. As demonstrated, RevGNN outperforms the GCN-Mean algorithm across all tasks, achieving an average nomination accuracy of 0.74. We will include these results in an updated version of our submission. Additionally, since RevGNN is now the best-performing model, we have re-run the GPT-4 refinement experiment using the RevGNN model in our response to reviewer fDbi. Thank you for bringing this model to our attention.
>
> | Model| cart btn  | name  | price  | buy btn  | main img  | subj | avg  |
> |-------------------|--------------|----------|-----------|-------------|--------------|----------|---------|
> | GCN- Mean 2 Layer | 0.62         | 0.78     | 0.66      | 0.78        | 0.61         | 0.84     | 0.71   |
> | REVGNN - 20 layer | 0.65 | 0.79 | 0.66 | 0.81 | 0.62 | 0.89 | 0.74 |
>
> We also attempted to train the requested GPS model based on the following [implementation](https://github.com/pyg-team/pytorch_geometric/blob/master/examples/graph_gps.py), using both [random walk positional encodings](https://pytorch-geometric.readthedocs.io/en/latest/generated/torch_geometric.transforms.AddRandomWalkPE.html) and [Laplacian eigenvector positional encodings](https://pytorch-geometric.readthedocs.io/en/latest/generated/torch_geometric.transforms.AddLaplacianEigenvectorPE.html#torch_geometric.transforms.AddLaplacianEigenvectorPE), but unfortunately could not obtain a satisfactory performance. As mentioned in Answer R4, we also explored positional encodings based on traversing the tree in depth-first and breadth-first orders, but found that including these encodings hurt performance. The lack of effectiveness of positional encodings in our context may be due to the significant variation in DOM tree structures across websites. In our dataset, the number of nodes can range from just a few hundred to around 20,000. However, a more consistent representation of page structure might be the locations of node elements in the corresponding rendered screenshot. For instance, a screenshot of a page on an iPhone X device will have a consistent width across websites, and the position of elements in the screenshot tends to remain stable (e.g., the "go to cart" button is usually in the upper right corner). We will mention the possibility of using positional encodings based on the screenshot in the future work section.
>
> We would also like to clarify why we did not include some of the requested models. In the past, we invested significant effort in training Graph Attention Networks on our dataset, using both a custom implementation and the pre-implemented PyTorch Geometric versions (GATv2Conv and GATConv). However, the GAT's performance was consistently inferior to the other models in our evaluation, which led us to exclude it from our paper. However, given the GAT's popularity, we recognize that other researchers will likely consider using it for similar tasks. To assist them, we will provide a detailed explanation in Section 5, "Embedders for DOM Elements," outlining the reasons for excluding the GAT from our study.
>
> Finally, we did not include the GIN model as it is primarily useful when the structural properties of the graph are what is central for solving the task at hand. In the context of web pages, the topology of the DOM-tree alone is insufficient for accurately locating web elements. The nodes in the DOM-tree carry a lot of additional information that is absolutely necessary for rendering the page and, consequently, for element nomination. This information includes attributes, bounding box dimensions, textual content, HTML tag types, x and y coordinates on the rendered page, style information, etc. Given the importance of the node features, we chose not to include GIN, as it does not leverage this additional, essential information.
>
> # R2
> Learning element embeddings using heterogeneous GNNs is, as far as we are aware, a novel and unexplored topic in this domain. Therefore, it is unclear to us how to model the DOM-trees as heterogeneous graphs.
>
> However, it is worth noting that the TreeLSTM models (LSTM-Bi, LSTM-BiE, LSTM-TD, and LSTM-BU) employed in our study do distinguish between parent and child nodes within their architectures. Despite this, the LSTMs are almost exclusively outperformed by the GCN-MEAN algorithm, which does not distinguish between these types of nodes.

---

> ### Author Response · Authors · 2024-08-26
>
> # R3
>
> Although training on the classification task and evaluating based on nomination is a common practice in the literature, we understand that this may not be immediately obvious to those unfamiliar with the field. Therefore, we will include a more thorough discussion of this in the Appendix, based on our response below.
>
> In the past we have designed and empirically evaluated numerous approaches for learning to perform element nomination along the lines you suggested but have not been able to formulate a nomination loss function which performs close to the classification objective. The main issue is how to formulate the cross entropy objective such that it assigns high mass to the correct node of the correct class and low mass to all other nodes.
>
> To understand why this is not straightforward, let’s say that we have $L$ class labels and that we for now disregard the unlabeled class. Thus, if we have $N$ nodes on a page, we could use $L$ one-hot vectors of length $N$ to encode which nodes belong to each class on a page.
>
> Imagine that we map the hidden representations of the $N$ nodes on the page into an $N \times L$ matrix using a function $f_\theta(\cdot)$ and subsequently apply softmax along the nodes for all $L$ classes. Then we could formulate a cross-entropy loss function as:
>
> \begin{equation}
>     \sum_{l=1}^{L} \sum_{n=1}^{N} p(x_{n,l}) \log q(x_{n,l}).
> \end{equation}
> After incorporating the one-hot encoded labels, this double sum just becomes a sum over $L $ terms, one for each class.
>
> The reason why this objective does not work is that it does not instruct $f_\theta(\cdot)$ how to properly handle unlabeled nodes. When $f_\theta(\cdot)$ maps a labeled node into the $L $ class probabilities, it simply assigns a high amount of mass to the $ l$-th correct class. However, for an unlabeled node, it should not learn to assign any mass to any of the $ L$ classes, yet because it is a mapping to these $ L $ classes it is forced to distribute its mass among these classes somehow.
>
> A natural suggestion for handling this issue is to incorporate the unlabeled class such that $f_\theta(\cdot)$ maps the node embeddings into an $N \times (L+1)$ matrix. Now it remains to decide how to represent which nodes belong to which class. For the labeled nodes, we keep the $L$ one-hot vectors of length $N$ to encode which nodes belong to each class. However, it is now unclear how to encode the unlabeled nodes. One approach could be to use a vector of $N - L$ ones and $L$ zeros, where $N - L$ corresponds to the number of unlabeled nodes on the page.
>
> The issue with this approach is related to class imbalance. After incorporating the labels into the equation above, we now have $L + (N-1)$ terms, where $L$ terms incentivize the model to identify the $L$ labeled nodes correctly, while $(N-L)$ terms (where $N$ can be up to $20,000$) incentivize the model to learn to recognize unlabeled nodes correctly. With this very large imbalance, the model might as well predict that everything is unlabeled.
>
> One suggestion to mitigate the class imbalance could be to uniformly weight the terms in the loss which correspond to the unlabeled nodes uniformly, resulting in the following cross-entropy formulation:
>
> $$
>     \sum_{l=1}^{L} \sum_{n=1}^{N} p(x_{n,l}) \log q(x_{n,l}) + \frac{1}{(N-L)}\sum_{n=1}^{N} p(x_{n,(L+1)}) \log q(x_{n,(L+1)}).
> $$
>
> The issue with this approach is that the model is now less incentivized to correctly identify every single unlabeled node. The implication is that when it encounters difficult negative examples, it is less punished for incorrectly assigning more mass to one of the $L$ labeled classes instead of the unlabeled class. Not learning to handle hard negative examples can significantly degrade the final nomination performance, as it just takes a single confusing element to ruin the nomination.
>
> # R4
> *In the tree setting, the depth would also seem to be a natural encoding* - We did explore using positional encodings by traversing the tree in both depth-first and breadth-first orders. However, we found that including these positional encodings negatively impacted the performance of the transformer model. We will mention this in the submission. However, you are correct that some of the positional encodings used in graph transformers could also be useful here, and we will mention this in the future work section.

---

> ### Author Response · Authors · 2024-08-26
>
> # R5
> To investigate the variability of the methods explored in this paper, we conducted the experiments presented in Table 2 with four additional initializations. The results are summarized in the Table below. In this table, we report the average performance over five runs, with the standard deviation shown in parentheses.
>
> The table demonstrates that the GCN-Mean algorithm consistently achieves the highest accuracy across most tasks, and in the few instances when it is not the best, it ranks among the best methods. Also, across all tasks, the GCN-Mean algorithm achieves the highest overall average accuracy by a significant margin. We can also deduce that the performance of the GCN-Mean algorithm is consistent. In contrast, larger models like LSTM-BU exhibit more variability and a higher variance.
>
> | Model          | buy button w/o text | buy button w/ text | cart button w/o text | cart button w/ text | main image w/o text | main image w/ text | name w/o text | name w/ text | price w/o text | price w/ text | subj. w/o text | subj. w/ text | AVG Accuracy w/o text | AVG Accuracy w/ text |
> |----------------|---------------------|--------------------|----------------------|---------------------|---------------------|--------------------|---------------|--------------|----------------|---------------|----------------|---------------|-----------------------|----------------------|
> | LSTM-Bi*       | .72                 | -                  | .57                  | -                   | .466                | -                  | .693          | -            | .483           | -             | .656           | -             | 0.598                 | -                    |
> | LSTM-BiE*      | .733                | -                  | .596                 | -                   | .503                | -                  | .724          | -            | .546           | -             | .702           | -             | 0.634                 | -                    |
> | GCN-GRU        | .767(.0097)         | .832(.0031)        | .569(.0045)          | .623(.003)          | .406(.0057)         | .376(.0041)        | .693(.0013)   | .791(.0033)  | .503(.021)     | .604(.0026)   | .451(.0032)    | .532(.0028)   | 0.56483               | 0.6263               |
> | FreeDOM**      | -                   | .293(.085)         | -                    | .056(.043)          | -                   | .043(.021)         | -             | .57(.068)    | -              | .226(.025)    | -              | .351(.015)    | -                     | 0.2565               |
> | FreeDOM-ext**  | -                   | .835(.012)         | -                    | .591(.0073)         | -                   | .312(.0092)        | -             | .836(.0099)  | -              | .721(.022)    | -              | .461(.024)    | -                     | 0.626                |
> | LSTM-TD        | .709(.019)          | .849(.036)         | .498(.021)           | .514(.0158)         | .418(.0175)         | .456(.023)         | .697(.011)    | .758(.00)    | .52(.0035)     | .582(.028)    | .589(.024)     | .34(.0105)    | 0.5718                | 0.58316              |
> | LSTM-BU        | .725(.06)           | .89(.034)          | .571(.03)            | .651(.021)          | .451(.031)          | .435(.023)         | .692(.021)    | .741(.010)   | .463(.064)     | .531(.0245)   | .656(.021)     | .561(.02)     | 0.593                 | 0.636                |
> | TE             | .693(.02)           | .812(.011)         | .433(.031)           | .458(.036)          | .252(.063)          | .344(.024)         | .731(.029)    | .787(.007)   | .602(.0251)    | .622(.093)    | .55(.03)       | .622(.093)    | 0.5435                | 0.6075               |
> | FCN            | .785(.007)          | .860(.0082)        | .525(.0039)          | .618(.0088)         | .315(.0293)         | .272(.0028)        | .701(.0095)   | .851(.0103)  | .583(.0862)    | .611(.012)    | .429(.04)      | .39(.01)      | 0.5563                | 0.6003               |
> | GCN-Mean       | .775(.0365)         | .914(.0038)        | .62(.0045)           | .674(.0252)         | .6108(.016)         | .481(.014)         | .782(.0039)   | .811(.002)   | .66(.0237)     | .657(.004)    | .841(.006)     | .857(.0021)   | 0.7148                | 0.7323
>
> *Note: Both FreeDOM and FreeDOM-ext incorporate text in the authors' original implementations.
>
> **These results are based on a single run due to the high computational expense of these models.

---

> ### Author Response · Authors · 2024-08-26
>
> # R6
>
> We have run the requested sensitivity analysis, varying the number of layers in the GCN on a subset of $8000$ pages. As observed, the GCN's performance initially improves but then degrades with the addition of a third layer, indicating the presence of over-smoothing and over-squashing. We will include these results in the Appedix of the Submission.
>
>
> |                | 1 Layer | 2 Layers | 3 Layers | 4 Layers |
> |----------------|---------|----------|----------|----------|
> | Avg Nomination | 0.64853 | 0.67898  | 0.66595  |   0.63130|

---

> > ### Comment · Reviewer_grL9 · 2024-09-01
> > **Changes to the paper?**
> >
> > Could the authors please indicate where changes have been made to the paper to address the reviewers' concerns?
> >
> > The responses in openreview are good and address my concerns for the most part, but it is not always clear how the authors have modified the actual paper to incorporate the reported results/discussion. When I look at the revised paper, I struggle to see exactly which sections have changed.
> >
> > Maybe I am downloading the wrong file, but I have tried accessing it both through the main page (where the latest version is usually available) and through the revisions section.  A version of the paper with changed text/sections in blue would be helpful.

---

> > > ### Author Response · Authors · 2024-09-01
> > >
> > > Dear Reviewer,
> > >
> > > We will upload a revised version of the submission tomorrow morning, following the "Anywhere on Earth" time zone. The changes we have made will be highlighted in blue.

---

> ### Author Response · Authors · 2024-09-02
>
> Dear Reviewer,
>
> We have submitted a revised version of the paper, incorporating the promised updates. Please let us know if you have any further comments or if you believe anything is missing from the revision.

---

### Review · Reviewer_Y8Jf · 2024-08-07

**Summary Of Contributions:**

This paper proposes a novel dataset for identifying and classifying elements on web pages from DOM trees.

Firstly, it is pitched as an interesting benchmark for graph representation learning, and a simple GCN baseline is shown to perform remarkably well compared to Transformer-style encoders, TreeLSTMs and prior art specialised for the DOM input.

Secondly, it is demonstrated how the outputs and predictions from the GCN can inform targeted prompts to a language model, which can further refine and boost the predictive accuracy.

**Audience:**

Yes

**Claims And Evidence:**

Yes

**Requested Changes:**

**Major**:

Please gather results across multiple runs of your baselines and use these to provide error bars in the paper.

**Minor**:

Please consider adding a more thorough discussion of _graph rewiring_ in your paper, and ideally, some baselines to that effect. Currently I saw no obvious discussions of potential benefits of changing the DOM graph structure; even though it is well known in the community that trees suffer from the over-squashing effect (see, e.g., Alon and Yahav, ICLR'21; Topping, Di Giovanni et al., ICLR'22, Di Giovanni et al., ICML'23, Di Giovanni, Rusch, et al., TMLR'24) which causes GNNs to be hard to learn in this regime.

The "Transformer" baseline discussed here, if I understand it correctly, would do a form of graph rewiring (as all nodes are fed together as token input for a Transformer); but it is, to the best of my knowledge, not setup like a typical Graph Transformer (Müller et al., TMLR'24), as e.g. you do not discuss any positional embeddings that could have been used.

And even then, Transformers do rewiring in a limited sense (fully connected / lower-triangular graph depending on whether the Encoder or Decoder is used). There are other interesting rewiring techniques; see e.g. DGC (Gasteiger et al., NeurIPS'19), SDRF (Topping, Di Giovanni et al., ICLR'22) or EGP (Deac et al., LoG'22). It would be very interesting to see if any benefits can be made if we go beyond a tree structure---both a positive and a negative result would be insightful.

**Minor**:

While I appreciate that an LLM using the ranking from a GCN can outperform a GCN alone, nothing is said about whether it outperforms the LLM alone :)

i.e., I see no clear evidence in the present paper that, if we just fed the LLM with all the possibilities (instead of the top-K ones) -- or even, the correct possibility and K randomly-sampled ones -- that it wouldn't perform well. It would probably be useful to provide that evidence (if computational constraints allow for it).

**Strengths And Weaknesses:**

While I am not an expert in the specific area of DOM tree analysis, my impression from reading the paper is that the dataset addresses an unmet need, and is constructed in an interesting and careful manner. For that alone, the paper should likely merit publication.

On top of this, the results are very interesting---especially the strong performance of the GCN with averaging. The fact its predictions are amplified when piping the top-k raw HTML into an LLM are not particularly surprising, but valuable to see nonetheless.

The main issue I take with the work as-is from the TMLR perspective is that there are absolutely no error bars or mentions of repeated runs / seeds in the paper (that I was able to see). Especially since this is a benchmark paper that would likely set standards for papers to follow, I find it absolutely important to provide confidence measures around the accuracy metrics provided.

Beyond this, I find that the proposed baselines either overly focus on the input graph structure (which is a tree, and hence known to be suboptimal for information propagation due to bottlenecks), or too little. Further, the aspect of LLM classifiers could have been better explored with further experiments. I will outline several ways in which the paper can be improved w.r.t. this aspect in the "Requested Changes" section.

---

> ### Author Response · Authors · 2024-08-26
>
> # Major requested change - Error bars
>
> To investigate the variability of the methods explored in this paper, we conducted the experiments presented in Table 2 with four additional initializations. The results are summarized in the Table below. In this table, we report the average performance over five runs, with the standard deviation shown in parentheses.
>
> The table demonstrates that the GCN-Mean algorithm consistently achieves the highest accuracy across most tasks, and in the few instances when it is not the best, it ranks among the best methods. Also, across all tasks, the GCN-Mean algorithm achieves the highest overall average accuracy by a significant margin. We can also deduce that the performance of the GCN-Mean algorithm is consistent. In contrast, larger models like LSTM-BU exhibit more variability and a higher variance.
>
> | Model          | buy button w/o text | buy button w/ text | cart button w/o text | cart button w/ text | main image w/o text | main image w/ text | name w/o text | name w/ text | price w/o text | price w/ text | subj. w/o text | subj. w/ text | AVG Accuracy w/o text | AVG Accuracy w/ text |
> |----------------|---------------------|--------------------|----------------------|---------------------|---------------------|--------------------|---------------|--------------|----------------|---------------|----------------|---------------|-----------------------|----------------------|
> | LSTM-Bi*       | .72                 | -                  | .57                  | -                   | .466                | -                  | .693          | -            | .483           | -             | .656           | -             | 0.598                 | -                    |
> | LSTM-BiE*      | .733                | -                  | .596                 | -                   | .503                | -                  | .724          | -            | .546           | -             | .702           | -             | 0.634                 | -                    |
> | GCN-GRU        | .767(.0097)         | .832(.0031)        | .569(.0045)          | .623(.003)          | .406(.0057)         | .376(.0041)        | .693(.0013)   | .791(.0033)  | .503(.021)     | .604(.0026)   | .451(.0032)    | .532(.0028)   | 0.56483               | 0.6263               |
> | FreeDOM**      | -                   | .293(.085)         | -                    | .056(.043)          | -                   | .043(.021)         | -             | .57(.068)    | -              | .226(.025)    | -              | .351(.015)    | -                     | 0.2565               |
> | FreeDOM-ext**  | -                   | .835(.012)         | -                    | .591(.0073)         | -                   | .312(.0092)        | -             | .836(.0099)  | -              | .721(.022)    | -              | .461(.024)    | -                     | 0.626                |
> | LSTM-TD        | .709(.019)          | .849(.036)         | .498(.021)           | .514(.0158)         | .418(.0175)         | .456(.023)         | .697(.011)    | .758(.00)    | .52(.0035)     | .582(.028)    | .589(.024)     | .34(.0105)    | 0.5718                | 0.58316              |
> | LSTM-BU        | .725(.06)           | .89(.034)          | .571(.03)            | .651(.021)          | .451(.031)          | .435(.023)         | .692(.021)    | .741(.010)   | .463(.064)     | .531(.0245)   | .656(.021)     | .561(.02)     | 0.593                 | 0.636                |
> | TE             | .693(.02)           | .812(.011)         | .433(.031)           | .458(.036)          | .252(.063)          | .344(.024)         | .731(.029)    | .787(.007)   | .602(.0251)    | .622(.093)    | .55(.03)       | .622(.093)    | 0.5435                | 0.6075               |
> | FCN            | .785(.007)          | .860(.0082)        | .525(.0039)          | .618(.0088)         | .315(.0293)         | .272(.0028)        | .701(.0095)   | .851(.0103)  | .583(.0862)    | .611(.012)    | .429(.04)      | .39(.01)      | 0.5563                | 0.6003               |
> | GCN-Mean       | .775(.0365)         | .914(.0038)        | .62(.0045)           | .674(.0252)         | .6108(.016)         | .481(.014)         | .782(.0039)   | .811(.002)   | .66(.0237)     | .657(.004)    | .841(.006)     | .857(.0021)   | 0.7148                | 0.7323
>
> *Note: Both FreeDOM and FreeDOM-ext incorporate text in the authors' original implementations.
>
> **These results are based on a single run due to the high computational expense of these models.

---

> ### Author Response · Authors · 2024-08-27
>
> # A More Thorough Discussion of Graph Rewiring
>
> ## *I have not found any comprehensive discussions on the potential benefits of altering the DOM graph structure.*
>
> We agree that graph rewiring could potentially be a valuable approach in the context of web element nomination. The sensitivity analysis experiment provided to reviewer grL9, where we gradually increased the number of layers, demonstrated that performance improved up to $2$ layers but then declined as more layers were added. This suggests that the GCN is indeed prone to over-squashing in this context. Based on these findings, we will include a discussion in the future work section about the potential benefits of modifying the DOM graph structure to mitigate these issues.
>
> Expander Graph Propagation, in particular, appears to be a highly promising and scalable graph rewiring approach for DOM trees. We will highlight it as a promising direction for future work in our submission and plan to explore its application in our domain moving forward. Thank you for bringing it to our attention.
>
> GDC might not be the most effective rewiring technique for web element nomination, as it is likely important to leverage information from both similar and dissimilar elements. Our understanding is that GDC is based on the principle of homophily, which suggests that similar nodes are more likely to be connected. By diffusing information across the graph, GDC tends to reinforce connections between these similar nodes. However, in the context of element nomination, valuable insights can be gained from both similar and dissimilar elements. For instance, when nominating a product price element, it is helpful to consider connections to similar price elements from recommended products on the page. Simultaneously, it can also be beneficial to include connections to dissimilar elements, such as the product title element.
>
> ## The "Transformer" Baseline - Graph Rewiring
> Yes, the transformer baseline does indeed implement a form of graph rewiring.
>
> ## The "Transformer" Baseline - No Discussion of Positional Embeddings
>
> We explored the use of positional encodings by traversing the DOM tree in both depth-first and breadth-first orders. However, we found that these positional encodings negatively impacted the transformer's performance, and we will mention this finding in the paper. In our response to reviewer grL9, we also explored using [random walk positional encodings](https://pytorch-geometric.readthedocs.io/en/latest/generated/torch_geometric.transforms.AddRandomWalkPE.html) and [Laplacian eigenvector positional encodings](https://pytorch-geometric.readthedocs.io/en/latest/generated/torch_geometric.transforms.AddLaplacianEigenvectorPE.html#torch_geometric.transforms.AddLaplacianEigenvectorPE) when training the GPS model. However, this model did not perform satisfactorily.
>
> The lack of effectiveness of positional encodings in our context may be due to the significant variation in DOM tree structures across websites. In our dataset, the number of nodes can range from just a few hundred to around 20,000. However, a more consistent representation of page structure might be the locations of node elements in the corresponding rendered screenshot. For instance, a screenshot of a page on an iPhone X device will have a consistent width across websites, and the position of elements in the screenshot tends to remain stable (e.g., the "go to cart" button is usually in the upper right corner). We will mention the possibility of using positional encodings based on the screenshot in the future work section.

---

> ### Author Response · Authors · 2024-08-27
>
> # While I appreciate that an LLM using the ranking from a GCN can outperform a GCN alone, nothing is mentioned about whether it outperforms the LLM alone
> ## Fed the LLM with all the possibilities
> We agree that it would be very interesting to see if an LLM alone could identify a specific element from all possible elements on a page. However, due to the limited context window and budget constraints, we unfortunately cannot input the large number of elements and even larger number of tokens into the LLM.
>
> ## The correct possibility and K randomly-sampled ones
> Usually, a trained model must compare every single element on a page to identify a specific target element. The purpose of using the GCN to pre-filter down to the top-K elements is to reduce the number of nodes the LLM needs to process. When filtering with the GCN, it is crucial to not filter out the true target node. Thus, the top-K elements identified by the GCN are those it considers most likely to be the target element. As a result, the LLM is presented with what the GCN regards as the K most challenging/confusing elements (hopefully including the correctly labeled element) on the page. In contrast, if we provide it with the correct element and K-1 randomly sampled nodes, we would be testing it in the easiest setting possible.

---

> > ### Comment · Reviewer_Y8Jf · 2024-08-28
> >
> > Thank you for your response and including the error bars -- this is now sufficient for me to recommend acceptance.
> >
> > I appreciate the infeasibility to feed the LLM all items at once due to context limitations.
> >
> > Perhaps instead the LLM could be prompted in a 'tournament' style? e.g. you cut your input into N/K parts with K elements in each, ask the LLM to choose the most likely one in each part, then do the same with the N/K winners, until only one element remains.

---

> > > ### Author Response · Authors · 2024-09-01
> > >
> > > Dear Reviewer,
> > >
> > > Thank you for the additional valuable input. The idea of using a 'tournament' style approach to prompt the LLM indeed seems promising for addressing the limitations of context size.
> > >
> > > However, an advantage of using GCNs for the top K selection is that it presents the LLM with a set of nodes that the GCN finds most challenging, which may differ from what the LLM finds difficult. In contrast, the tournament-style approach requires the LLM to compare elements that the LLM itself finds most challenging. This suggests there could be benefits to employing a different model for pre-filtering.

---

### Review · Reviewer_fDbi · 2024-08-13

**Summary Of Contributions:**

The authors present a couple of contributions in their manuscript. More specifically, they:
- introduce the *Anonymised Product Page Dataset*, a large-scale dataset of 51,701 manually labeled product pages from 8,175 e-commerce websites across 8 geographic regions. Based on the authors' comments, the design motivation is web element nomination tasks and includes labels for action elements (buy/cart buttons) and information elements (price, name, image).

- benchmark various GNNs on the web element nomination task using this dataset, including recurrent GNNs (Tree LSTMs), convolutional GNNs (GCNs), and attention-based GNNs.

- show that a simple 2-layer Graph Convolutional Network (GCN-Mean) outperforms more complex state-of-the-art methods like FreeDOM and DOM-Q-NET by a significant margin.
- show a post-training nomination refinement step using GPT-4, which further improves nomination accuracy by 16.82 percentage points on average.
- propose a novel Challenge Nomination Training Procedure (CNTP) to bridge the gap between classification and nomination objectives, improving nomination accuracy by about 5 percentage points.
- correlate the impact of textual features on nomination accuracy, showing they can improve performance for certain tasks but may be detrimental for others.

**Audience:**

Yes

**Broader Impact Concerns:**

I do not see a specific requirement for a broader impact statement section.

**Claims And Evidence:**

Yes

**Requested Changes:**

Based on the mentioned weaknesses, I identify the next changes as some useful changes for improving the quality of the work.
1. Expand the GPT-4 refinement experiment to a larger subset of the test set for more robust results.
2. Experiment on a subset of more recent web pages to validate the methods' effectiveness on current web designs.
3. Would the potential of combining the GCN-Mean approach with other high-performing methods like LSTM-BiE make sense?
4. Probably include visualizations of example nominations to provide intuition for the models' behavior.

**Strengths And Weaknesses:**

**Strengths**

1. The Anonymised Product Page Dataset itself is a strength, addressing the lack of large-scale, realistic datasets for web automation tasks. Its diversity across websites and languages could be particularly valuable.
2. The evaluation of various GNN architectures seems to be satisfying with respect to showing their performance in a task such as web element nomination tasks.
3. *Efficiency finding:* a simple GCN can outperforms more complex methods on web data.
4. The proposed CNTP and GPT-4 refinement step show that it could be a promising direction for improveming the nomination accuracy.

**Weaknesses**
Overall, I find the paper very interesting. However, I identify a couple of minor/moderate weaknesses that could help make the work more complete. More specifically:
1. The evaluation is limited to product pages. I wonder whether such an evaluation could generalize to other types of web pages or tasks.
2. The dataset, while large, is from 2018-2019 and may not reflect current web design trends.
3. The GPT-4 refinement experiment is limited to 100 pages due to budget constraints, which may not be fully representative. Based on the initial positive observations, it'd be great to investigate how GPT-4 further refinement can improve the model.

---

> ### Author Response · Authors · 2024-08-26
>
> # 1. Expand the GPT-4 refinement experiment
>
> Below are the results of the GPT-4 refinement experiment, where we increased the test set size from $100$ to $500$ pages. Since the 20-layer RevGNN model requested by reviewer grL9 outperformed the GCN-mean algorithm, we chose to compare against this stronger model instead. The results from this experiment can be found in the table below. From the table, we see that the performance of the LLM remains similar to the evaluation run on the smaller test set. Given that RevGNN is a stronger model than GCN, the performance gains from the refinement step have understandably decreased; however, the improvement remains substantial at $10.9$ percentage points.
>
> |       | RevGNN  | GPT 4  |
> |-------|---------|--------|
> | cart btn     | 0.791   | 0.887  |
> |  name    | 0.762   | 0.885  |
> | price     | 0.725   | 0.808  |
> | buy btn     | 0.872   | 0.976  |
> | image     | 0.522   | 0.661  |
> | **AVG** | **0.734** | **0.844** |
> | **Diff** | **0.109** |        |
>
> # 2. Experiment on more recent web pages
>
> We have previously considered this potential issue and, to address it, collected a test set of pages from 2020 to 2023. Validation on these more recent pages revealed no significant performance differences compared to the original dataset used in our submission. This suggests that e-commerce design patterns have remained relatively stable over the past few years.
>
> # 3. Would the potential of combining the GCN-Mean approach with other high-performing methods like LSTM-BiE make sense?
>
> Yes - Despite its promising performance, the LSTM-BiE model is constrained by high computational costs due to the potentially thousands of nodes in the DOM tree. To address this complexity, one potential solution is to use a pre-trained GCN-Mean algorithm to identify informative and non-informative nodes. Less relevant nodes adjacent to more informative ones can then be collapsed into the more informative neighbors. Collapsing nodes involves merging or combining them into a single node to simplify the tree structure. By simplifying the tree structure in this way, the LSTM-BiE model can be trained on a more manageable, reduced tree, thereby improving computational efficiency.
>
> # 4. Visualizations of example nominations to provide intuition for the models' behavior
> We have uploaded a revised version of the submission, which includes visualizations of instances where the RevGNN algorithm fails. Please refer to Figure 10 in the appendix.

---

> ### Author Response · Authors · 2024-08-27
>
> Dear Reviewer fDbi,
>
> We have now uploaded the revised version of the submission, which includes visualizations of where the RevGNN algorithm fails, as shown in Figure 10 in the appendix. We apologize for the delay.

---

### Author Response · Authors · 2024-08-28

Dear Reviewers,

Thank you for the valuable time you invested in understanding and reviewing our work.

In response to your suggestions, we have re-run the experiments presented in Table 2 using five different initializations, underscoring the robustness of the GNNs. We also expanded the GPT-4 refinement experiment to include a larger test subset, further strengthening our conclusions regarding the usefulness of LLMs for element nomination.

Additionally, we have incorporated the state-of-the-art RevGNN architecture into our analysis. Our results show that RevGNN outperforms the previously best-performing 2-layer GCN-Mean method, reinforcing our claims that GNNs hold untapped potential for web element nomination. This also suggests that even more effective GNNs are likely yet to be discovered, and thus GNNs deserve greater attention from practitioners in web automation.

Below, we have responded in detail to each of your individual comments. Once again, thank you for your valuable feedback.

---

### Decision · Action_Editor_Wem9 · 2024-10-05

**Recommendation:** Accept as is

**Comment:**

The paper's introduction of a valuable dataset, combined with its rigorous benchmarking makes a contribution to web automation research.

The reviewers had generally positive feedback, and recommended accepting it for publication, although with some suggestions for improvement.
grL9: "The paper introduces an impressive new dataset and explains how it can be used for the web element nomination task. The paper makes a convincing case that this task is practically important, but research in the field suffers from a lack of large public datasets that cover multiple markets and languages."

**Audience:**

The paper is of interest to the researchers working in the field of web automation, specifically those interested in web element nomination and related tasks, as well as those working on the GNNs.

**Claims And Evidence:**

The paper introduces a comprehensive dataset for web element nomination task and benchmark several Graph Neural Network architectures, finding that a simple Convolutional GNN outperforms other more complex state-of-the art methods. The authors also show that using a large language model to refine the results of the GNN leads to significant performance gains.